# Learning Transferable Adversarial Perturbations

**Krishna Kanth Nakka**[1], **Mathieu Salzmann**[1,2]
[1]CVLab, EPFL, Switzerland
[2]ClearSpace, Switzerland
{krishna.nakka, mathieu.salzmann}@epfl.ch

## Abstract

While effective, deep neural networks (DNNs) are vulnerable to adversarial attacks. In particular, recent work has shown that such attacks could be generated by another deep network, leading to significant speedups over optimization-based perturbations. However, the ability of such generative methods to generalize to different test-time situations has not been systematically studied. In this paper, we therefore investigate the transferability of generated perturbations when the conditions at inference time differ from the training ones in terms of target architecture, target data, and target task. Specifically, we identify the mid-level features extracted by the intermediate layers of DNNs as common ground across different architectures, datasets, and tasks. This lets us introduce a loss function based on such mid-level features to learn an effective, transferable perturbation generator. Our experiments demonstrate that our approach outperforms the state-of-the-art universal and transferable attack strategies.

## 1 Introduction

In recent years, deep neural networks (DNNs) have achieved great success in a wide range of applications [1, 2, 3, 4]. However, DNNs have been demonstrated to be vulnerable to adversarial examples [5] crafted by adding imperceptible perturbations to clean images. In particular, two broad categories of attacks have been studied. The first one consists of iterative algorithms [5, 6, 7], which optimize the perturbation for each instance, and thus tend to be computationally expensive. The second one encompasses generative methods [8, 9, 10], which train a deep network to produce perturbations. As such, attacking the target network only involves a forward pass through the generator, typically resulting in much faster attacks than iterative methods. However, speed is not the only important factor to assess the strength of an attacker; its ability to generalize to different situations is also key to its success.

In this paper, we therefore study the transferability of perturbations obtained with generative methods. Specifically, we investigate the transfer of such perturbations when the conditions at inference time differ from the training ones in terms of **(i)** target *architecture*, e.g., the generator was trained to attack a VGG-16 but the target network is a ResNet152; **(ii)** target *data*, e.g., the generator was trained using the Paintings dataset but the test data comes from ImageNet; **(iii)** target *task*, e.g., the generator was trained to attack an image recognition model but faces an object detector at test time. To the best of our knowledge, for generative methods, our work constitutes the first attempt at transferability across tasks, and only [11] has studied generalization across architectures and data, by introducing a loss function acting on the relative class probabilities of the attacked and unattacked examples.

Here, by contrast, we improve the transferability of a perturbation generator across architectures, data, and tasks by exploiting the mid-level features of DNNs. The key motivation behind our approach is our observation that the mid-level features extracted by DNNs with different architectures, different data, or for different tasks bear strong similarities. This is illustrated in Figure 1, where we visualize the features extracted by different backbones and for different tasks, and thus different datasets, using

35th Conference on Neural Information Processing Systems (NeurIPS 2021).

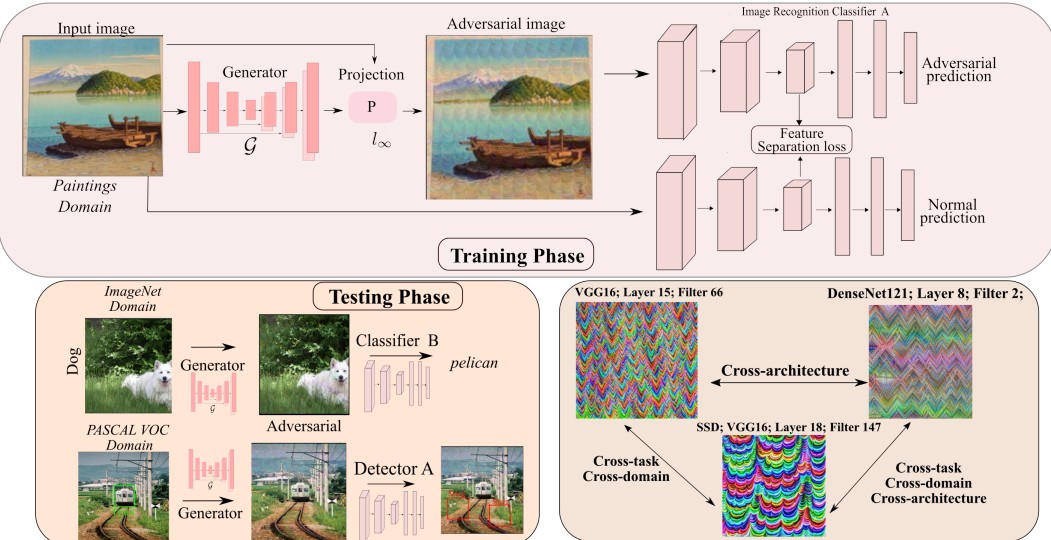

Figure 1: **Learning transferable perturbations.** We observe that mid-level features are common across architectures and tasks, and thus propose to exploit them to train a perturbation generator by maximizing the relative distance between normal and perturbed features. We show that such a generator is effective even in the presence of a different model, dataset, or task at test time.

the method of [12]. Our analysis suggests that adversarial perturbations that significantly affect the mid-level features of one sample in one architecture also affect them in a different architecture, even for a different task and with different data.

We therefore propose to train a perturbation generator by maximizing the distance between the normal features of a sample and their adversarial counterparts extracted in the intermediate layers of a pretrained classifier. The resulting perturbations are then transferable across architectures, datasets and tasks because they similarly affect the mid-level features of the corresponding filters in the target setup. The perturbed mid-level features are then propagated to the top layers of the network, and, as a result, lead to incorrect predictions.

**Contributions.** Our contributions can be summarized as follows: 1. We identify the intermediate features of CNNs as common ground across different architectures, different data distributions and different tasks. 2. We introduce an approach that exploits such features to learn an effective, transferable perturbation generator. 3. We systematically investigate the effect of target architecture and target data distribution on the transferability of adversarial attacks. Our experiments demonstrate that our approach yields higher fooling rates than the state-of-the-art universal [8] and transferable [11] attacks. Our code is available at `https://github.com/krishnakanthnakka/Transferable_Perturbations`.

## 2   Related Work

Below, we review the recent works on adversarial attacks, with a focus on generator-based approaches.

**Adversarial Attacks.** Adversarial attacks were first investigated in [5] to identify the vulnerability of modern deep networks to imperceptible perturbations in the context of image classification. Since then, several attack strategies have been studied, including single-step fast gradient descent [6, 13], and computationally more expensive optimization-based attacks, such as CW [14], JSMA [15], and others [16, 17, 7].

While the above methods are image-dependent, the existence of Universal Adversarial Perturbation (UAP) was first shown in [18], considering the task of learning a single perturbation that can fool a classifier independently of the input image. Such a UAP was iteratively updated based on its effectiveness to move the individual data samples across the decision boundary. Other UAPs based on the computation of the singular vectors of the Jacobian matrices of feature maps have been studied

in [19]. In parallel to UAPs, several works have shown that iterative adversarial attack strategies could be transferred across architectures [20, 21, 22]. In particular, the recent work of [23] proposed to improve targeted attack transferability across architectures by training auxiliary classifiers at each intermediate layer to attack a CNN's feature maps. The above-mentioned methods, however, either are computationally expensive at inference time [16, 7, 15, 14, 21, 23], or suffer from low transferability rates [18, 20, 22]. Furthermore, most of the attacks aiming for transferability [18, 19, 23] strongly depend on the availability of data from the target domain. By contrast, we introduce an efficient generative method to produce adversarial perturbations that generalize not only across architectures, but also across training data and across tasks.

**Generator-based Attacks.** Generative adversarial perturbations (GAP) were first introduced in [10]. In particular, [10] showed that a generator network can be used to craft a UAP that transform an input image to an image-dependent perturbation. Similarly, [24] introduced the advGAN generative framework to learn to produce adversarial perturbations, and further proposed to make use of distillation to perform black-box attacks. Based on the observation that the effectiveness of these two methods strongly depends on the availability of data from the target domain during training, [11] introduced the relativistic cross-entropy loss, which was shown to better generalize across datasets. Furthermore, [8] proposed to generate UAPs by transforming the generator's output using a regional norm layer that enforces perturbation homogeneity. While effective, the above-mentioned works explicitly rely on the classification boundary of the attacked model, which we will show to tend to make them overfit to the source data. By contrast, we identify the mid-level features as a more robust signal shared across not only different architectures and datasets, but even across different tasks. Our experiments will showcase the superiority of exploiting this information over the losses used in previous approaches to train a transferrable perturbation generator.

# 3 Methodolgy

Let $\mathbf{x}_i \in \mathbf{R}^{H \times W \times 3}$ be a color image of size $H \times W$, and $\mathbf{y}_i$ be the associated ground-truth label. Furthermore, let $f$ denote the task-related convolutional neural network that takes $\mathbf{x}$ as input and extracts, at layer $l = \{l_j\}_{j=1}^L$, a feature map $f_\ell(\mathbf{x}_i) \in \mathbf{R}^{N_\ell \times D_\ell}$, which we assume to be reshaped in matrix form, such that $N_\ell = H_\ell \times W_\ell$, with $H_\ell$ and $W_\ell$ the spatial dimensions of the feature map, and $D_\ell$ its number of channels.

Our goal is to train a generator $\mathcal{G}$ that produces a perturbation $\delta_i \in \mathbf{R}^{H \times W \times 3}$, which, when added to the clean image $\mathbf{x}_i$, results in predicting a different label from $\mathbf{y}_i$. To this end, we feed the input image $\mathbf{x}_i$ to the generator to synthesize an unbounded adversarial image $\mathcal{G}(\mathbf{x}_i)$, which is then clipped to be within an $\epsilon$ bound of $\mathbf{x}_i$ under the $\ell_\infty$ norm. Let $\hat{\mathbf{x}}_i$ be the adversarial image obtained after such a clipping of $\mathcal{G}(\mathbf{x}_i)$. In contrast to [10, 11], which rely on the final classification boundary of the task-related network $f$ to train the generator with a cross-entropy-based loss, we exploit the mid-level features of $f$. Specifically, we maximize the $L_2$ distance between the normal feature map $f_\ell(\mathbf{x}_i)$ and adversarial feature map $f_\ell(\hat{\mathbf{x}}_i)$ at layer $l$ using a feature separation loss term defined as

$$\mathcal{L}_{feat}(\mathbf{x}_i, \hat{\mathbf{x}}_i) = ||f_\ell(\mathbf{x}_i) - f_\ell(\hat{\mathbf{x}}_i)||_F^2 , \tag{1}$$

where $\| \cdot \|_F$ denotes the Frobenius norm.

The overall training scheme of our perturbation generator is provided in Algorithm 1. In practice, we observe the feature separation loss of Eq. 1 to encourage the generator to learn low-level, quasi-imperceptible structured patterns that affect that activations of a few filters while either suppressing the other ones or leaving them unaffected. More importantly, these perturbations are highly transferable.

**Reason for transferability.** Given the simplicity of our formulation, a natural question that arises is what affects the transferability of perturbations among CNNs. To understand this, we analyze the internal workings of a CNN, whose filters in each layer encode different levels of information. In particular, in Figure 2, we visualize the filters of various CNNs, sampled at different layers, using the technique of [12], which maximizes the mean activation of the each filter. This allows us to discover common characteristics across the different architectures. On one hand, the bottom filters, close to the input image, extract color and edge information, as shown in the left block of Figure 2. On the other, the top-level filters, close to the output layer, shown in the right block, are more focused on the object representation, and thus more task specific. By contrast, the mid-level filters learn more nuanced features, such as textures, and therefore tend to display similar patterns across architectures,

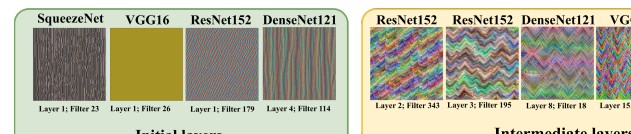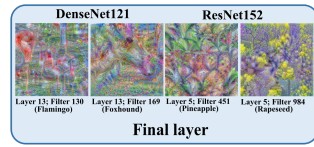

Figure 2: **Filter visualization.** The filters in the mid-level layers of different CNNs follow similar activation patterns, also common across tasks, whereas those near the classification layer focus on object-level features and are thus more task-specific.

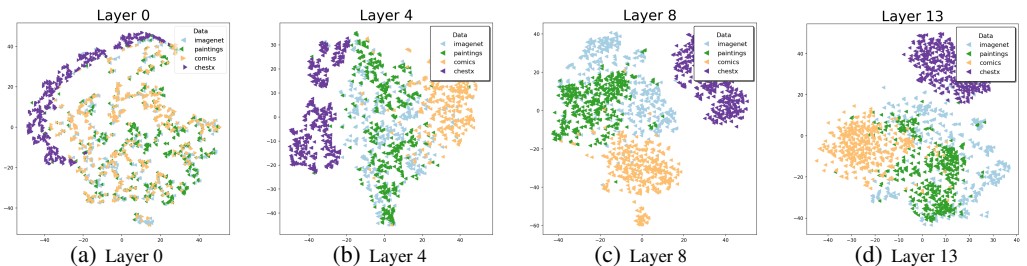

(a) Layer 0 (b) Layer 4 (c) Layer 8 (d) Layer 13

Figure 3: **Feature visualization.** We visualize the features extracted at 4 different layers of DenseNet121 for different datasets using t-SNE.

datasets tasks. As a consequence, and as indicated by our experiments, attacking the initial layers requires large perturbation strengths. Furthermore, while attacking the top-level features works well in the white box scenario, it does not transfer well to different data or architectures, because the features are already too architecture- and task-specific. Attacking mid-level features thus comes as a natural choice, which we will show to yield highly transferable perturbations.

Let us now focus on understanding the reason for the transferability when the target data is unavailable to the attacker. In Figure 3, we plot t-SNE visualizations of different datasets at 4 layers of DenseNet121. The proximity of the features extracted from the Comics or Paintings datasets, containing images from a few classes particularly focused on humans, to those extracted from ImageNet suggests these datasets as good candidates to train a generator that will fool an ImageNet-trained network (as shown in Tables 1 and 2). By contrast, the ChestX dataset presents a larger domain gap. Nevertheless, we will show that, albeit for a drop in fooling rate, our use of mid-level features still make a ChestX-trained generator reasonably effective on ImageNet-based classifiers. Furthermore, in Figure 4, we plot the top 80 activations of ImageNet, Paintings and ChestX at layer 8 of DenseNet121. This shows that the peaks of the ImageNet dataset (orange) are more strongly correlated to those of the Paintings dataset (left) than to those of ChestX (right). This is confirmed by our experiments, where a Paintings-trained generator is more effective than a ChestX-trained one. In short, while mid-level features lead to generator transferability, the resulting effectiveness remains affected by the similarity of the data.

---

**Algorithm 1** Training a transferable adversarial perturbation generator

---

**Input:** $\mathbf{x}_i$: clean images; $f$: pretrained classifer; $\epsilon$: perturbation $\ell_\infty$ bound

---

Initialize generator $\mathcal{G}$; Load classifier $f$ and freeze its parameters;
**repeat**
    Get a clean image $\mathbf{x}_i$ and feed $\mathbf{x}_i$ to $f$ to obtain $f_\ell(\mathbf{x}_i)$;
    Generate an unbounded adversarial image $\mathcal{G}(\mathbf{x}_i)$ and clip it within an $\epsilon$ bound of $\mathbf{x}_i$ to get $\hat{\mathbf{x}}_i$;
    Pass the adversarial image to $f$ to obtain $f_\ell(\hat{\mathbf{x}}_i)$;
    Compute the feature separation loss $\mathcal{L}_{feat}$;
    Compute the gradient of $\mathcal{L}_{feat}$ w.r.t. the weights of $\mathcal{G}$ and update these weights using Adam.
**until** convergence
**return** trained Generator $\mathcal{G}^*$

---

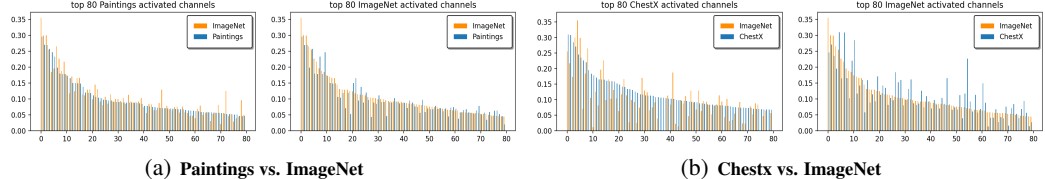

| | | top 80 Paintings activated channels | top 80 ImageNet activated channels | top 80 ChestX activated channels | top 80 ImageNet activated channels |
|---|---|---|---|---|---|

(a) **Paintings vs. ImageNet**  (b) **Chestx vs. ImageNet**

Figure 4: **Top activated channels.** Top-80 activated channels for ImageNet (orange) in comparison to Paintings in **(a)**, and to ChestX in **(b)**, for layer 8 of DenseNet121 using 500 images from each dataset. The blue bars are more correlated to the orange ones in the Paintings case, thereby achieving higher transferability rates than using ChestX. Best viewed in color and zoomed in.

| Gen. Training (data) | Discriminator | VGG16 | ResNet152 | Inception-v3 | DenseNet121 | SqueezeNet1.1 | Average (Black-box) |
|---|---|---|---|---|---|---|---|
| | | GAP [10] / CDA [11] / Ours | | | | | |
| ImageNet (1.2M) | VGG-16 | 99.9* / 99.8* / 99.3* | 53.5 / 53.6 / **68.4** | 41.7 / 43.2 / **46.6** | 58.9 / 66.5 / **84.7** | 67.8 / 70.6 / **86.5** | 55.5 / 58.5 / **71.6** |
| | ResNet152 | 93.2 / 96.8 / **99.1** | 97.6* / 99.6* / 99.7* | 60.5 / 66.0 / **74.9** | 87.5 / 94.2 / **98.8** | 83.9 / 82.8 / **89.1** | 81.3 / 84.9 / **90.5** |
| | Inception-v3 | 88.2 / 97.2 / **98.7** | 83.4 / 82.7 / **90.2** | 96.5* / 98.7* / 99.5* | 89.5 / 93.6 / **96.0** | 90.9 / 92.0 / **91.5** | 88.0 / 91.4 / **94.1** |
| | DenseNet121 | 94.9 / 95.0 / **99.4** | 89.5 / 91.0 / **98.7** | 56.1 / 57.7 / **86.0** | 99.6* / 99.6* / 99.6* | 79.7 / 81.5 / **95.6** | 80.1 / 81.3 / **94.9** |
| | SqueezeNet | 88.0 / 91.5 / **96.1** | 50.4 / 57.1 / **76.4** | 48.0 / 47.6 / **70.7** | 64.0 / 69.0 / **88.7** | 99.8* / 99.7* / 99.7* | 62.6 / 66.3 / **83.0** |
| | Average | 92.8 / 96.1 / **98.6** | 74.9 / 76.8 / **86.7** | 60.6 / 62.6 / **75.6** | 79.9 / 84.6 / **93.7** | 84.4 / 85.3 / **92.5** | 73.5 / 76.5 / **86.8** |

Table 1: **White-box and standard black-box settings.** We set $\epsilon = 10$ and report the fooling rate (in %) on 5K samples from the ImageNet val-set. The best result in each section is shown in bold. ∗ denotes the white-box setting.

## 4 Experiments

We evaluate the effectiveness of our attack strategy in diverse settings. Below, we first discuss our experimental setup and then compare our attacks with the state-of-the-art generator-based ones: CDA [11], RHP [8], and GAP [10]. To further demonstrate the generalizability of our approach, we also report its performance on adversarially-trained models and on the SSD [25] object detector with 4 different backbones. We discuss the experimental details in the supplementary material.

**Datasets.** To train the generator, similarly to [11], we use data from either ImageNet [26], Comics [27], Paintings [28] or ChestX [29] as source domain, containing 1.2M, 40K, 80K, and 8K images, respectively. We then randomly select 5000 images from the ImageNet [26] validation set as target domain to evaluate the transferability of our attacks. As in [11, 18, 10], we report the fooling rate and the absolute difference in top-1 error between before and after the attack. The fooling rate is the percentage of images for which the label is changed after the attack.

**Models.** For the target models, we make use of the publicly available PyTorch [30] versions of VGG-16 [31], VGG-19 [31], ResNet152 [32], DenseNet121 [33], Inception-V3 [34] and Squeezenet [35] pretrained on ImageNet. We chose this family of networks to understand the deeper impact of transferability across diverse architectures. As in [11], to evaluate transfer across datasets, we use ChestXNet [29] pretrained on ChestX. For the perturbation generators, we use the same architectures as in [11, 10]. To train them, we use the Adam optimizer with a learning rate of $2e^{-4}$ and a batch size of 16.

**Attack settings.** We perform attacks in four settings: **1.** White-box attacks, where the attacker has access to the exact target model and target data distribution; **2.** Standard black-box attacks, where the attacker has access to a substitute model from a different family of architectures trained on the target data and to the target data itself; **3.** Strict back-box attacks, where the attacker also uses a substitute model trained on the target data, but *without* having access to the target data itself; **4.** Extreme black-box scenario, we perform attacks without any knowledge of the target model or the target data.

### 4.1 Transferability to Unknown Target Model

Let us first study the standard white box and black-box settings. In this experiment, we train the generator with target data containing 1.2M ImageNet training samples, using one of the 5 ImageNet pretrained models as target architecture. We then evaluate the transferability of the attack to the remaining 4 models.

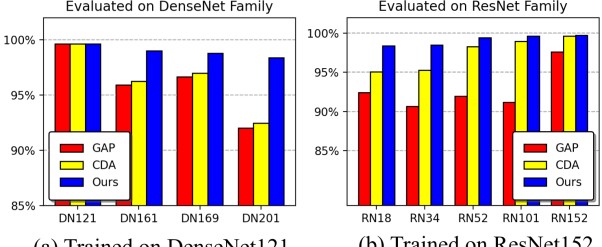

Figure 5: **Transferability within the same family.** We report the fooling rates when transferring attacks to other networks within the same family. The generators were trained with DenseNet121 and ResNet152, with $\epsilon = 10$.

(a) Trained on DenseNet121          (b) Trained on ResNet152

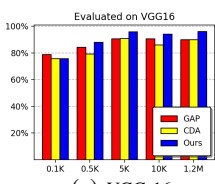 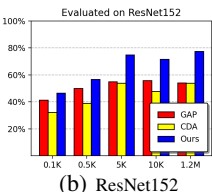 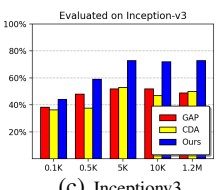 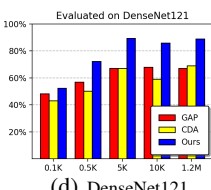

(a) VGG-16          (b) ResNet152          (c) Inceptionv3          (d) DenseNet121

Figure 6: **Limited data setup.** We report the fooling rates obtained using varying amounts of ImageNet training data. The generators were trained with SqueezeNet [35] and $\epsilon = 10$, and transferred to other networks. The $x$-axis represents the amount of training data.

Table 1 compares the effectiveness of our attacks to the CDA and GAP ones. We outperform these baselines on average by 10.5 and 13.5 percentage points (pp) (i.e., absolute difference of fooling rates), respectively. The differences are particularly pronounced when the generator is trained with SqueezeNet. Furthermore, in the DenseNet121 case, the difference between CDA and Ours is 13.6pp.

A similar trend can be seen when transferring attacks within the same family of networks. For instance, Figure 5 (a) shows the transferability when the generator is trained with DenseNet121 and evaluated on other networks of the same family, namely, DenseNet161, DenseNet169, and DenseNet201. Figure 5 (b) shows the results for similar experiments on the ResNet family. Our method performs on average 1.68pp, 6.4pp better than CDA and GAP in the case of ResNets, which demonstrates that our feature separation loss generalizes well for attacks within the same family. Importantly, the performance gap between Ours and the baselines increases with the difference in depth from training – from 0.02pp on DenseNet121 to 6.3pp on DenseNet201 w.r.t. GAP.

In Figure 6, we further study the transferability of attacks as the number of samples from the target data to train the generator varies. Our method (blue bar) consistently maintains higher fooling rates, even for a small number of training samples, than GAP (red bar) and CDA (yellow bar). Note that all methods tend to saturate when around 5K samples are available, with our method consistently outperforming the baselines.

## 4.2   Transferability to Unknown Target Data

Let us now turn to the strict black-box setting, where the target data is unavailable to the attacker, who then relies on a different dataset to train the generator. Note that, here, the attacker still has access to a substitute model trained on the target data, therefore we also study the impact of transferability to an unknown target model. Following [11], we consider two synthetic datasets, Comics and Paintings, depicting objects similar to the ImageNet ones, and one dataset, ChestX, containing a completely different type of data, thus suffering from a large domain gap.

Table 2 compares the results of our approach with those of the baselines. Our method yields clearly superior results for all 3 datasets. For instance, with Comics, Ours outperforms GAP and CDA on average by 14.9pp and 10.3pp, respectively, and with Paintings, by 14.9pp and 11.0pp, respectively. Note that, while we still outperform the baselines when using the ChestX data, by around 22pp on average, the fooling rates of all methods drop significantly in this challenging scenario. Nevertheless, altogether, these results evidence that one can learn to generate adversarial examples with high fooling rates despite not having access to the target data, even by using data from a completely

| Gen. Training (data) | Discriminator ( ImageNet) | VGG16 | ResNet152 | Inception-v3 | DenseNet121 | SqueezeNet 1.1 | Average |
|---|---|---|---|---|---|---|---|
| | | GAP [10] / CDA [11] / Ours | | | | | |
| Comics (40K) | VGG-16 | 99.8 / **99.9** / 99.5 | 54.3 / 54.0 / **77.4** | 45.8 / 45.2 / **61.9** | 66.3 / 64.2 / **93.6** | 70.7 / 68.4 / **93.4** | 67.4 / 66.3 / **85.1** |
| | ResNet152 | 75.3 / 95.8 / **99.3** | 97.6 / 98.1 / **99.6** | 31.7 / 66.5 / **73.1** | 45.1 / 87.7 / **98.6** | 67.3 / 86.0 / **90.7** | 63.4 / 86.8 / **92.3** |
| | Inception V3 | 84.3 / 85.6 / **99.0** | 97.2 / 97.3 / **90.4** | 99.8 / **99.8** / 99.6 | 88.5 / 88.0 / **96.7** | 82.4 / 82.3 / **93.2** | 90.5 / 90.6 / **95.8** |
| | DenseNet121 | 96.9 / 92.0 / **96.5** | **98.0** / 86.3 / 93.0 | **83.1** / 65.7 / 82.5 | 99.4 / 98.4 / **98.8** | 78.3 / 75.7 / **91.9** | 91.2 / 83.6 / **92.5** |
| | SqueezeNet | 87.7 / 89.9 / **96.5** | 54.0 / 58.2 / **79.0** | 51.2 / 51.4 / **75.4** | 68.7 / 76.3 / **90.2** | 99.7 / **99.8** / 99.7 | 72.3 / 75.1 / **88.2** |
| | Average | 88.8 / 92.6 / **98.2** | 80.2 / 72.8 / **87.8** | 62.3 / 65.8 / **78.5** | 73.6 / 82.9 / **95.6** | 79.7 / 82.5 / **93.8** | 76.9 / 80.5 / **90.8** |
| Paintings (80K) | VGG-16 | 99.4 / **99.9** / 99.0 | 41.1 / 57.6 / **66.6** | 36.5 / 46.6 / **50.0** | 50.8 / 73.8 / **84.6** | 63.7 / 73.0 / **86.4** | 58.3 / 70.1 / **77.3** |
| | ResNet152 | 80.4 / 89.9 / **98.7** | 95.4 / **97.5** / 99.4 | 50.7 / 62.1 / **72.8** | 70.4 / 82.3 / **97.9** | 70.4 / 81.1 / **89.2** | 73.5 / 82.6 / **91.6** |
| | Inception V3 | 80.3 / 80.5 / **98.6** | 95.8 / **96.4** / 88.2 | 99.6 / **99.6** / 99.5 | 87.7 / 87.2 / **95.2** | 77.5 / 72.8 / **90.8** | 88.2 / 87.3 / **94.5** |
| | DenseNet121 | 87.6 / 86.5 / **96.2** | 80.1 / 81.2 / **90.9** | 51.4 / 50.4 / **76.0** | 98.8 / **98.9** / 97.4 | 73.6 / 73.7 / **91.7** | 78.3 / 78.1 / **90.5** |
| | SqueezeNet | 82.8 / 80.7 / **95.2** | 46.0 / 46.0 / **73.4** | 44.5 / 47.4 / **71.0** | 59.3 / 56.5 / **87.2** | 99.4 / 99.3 / **99.6** | 66.4 / 66.0 / **85.3** |
| | Average | 86.1 / 87.5 / **97.6** | 71.7 / 75.8 / **83.7** | 56.5 / 61.2 / **73.9** | 73.4 / 79.7 / **92.5** | 76.9 / 80.0 / **91.5** | 72.9 / 76.8 / **87.8** |
| ChestX (10K) | VGG-16 | 78.7 / 85.6 / **93.3** | 23.2 / 23.3 / **41.8** | 25.5 / 27.9 / **31.3** | 27.5 / 28.2 / **53.4** | 46.1 / 48.0 / **64.3** | 40.2 / 42.6 / **56.8** |
| | ResNet152 | 39.9 / 44.8 / **56.4** | 27.0 / 25.3 / **62.8** | 28.2 / 25.7 / 27.7 | 25.9 / 26.6 / **38.1** | 44.9 / 47.1 / **60.5** | 33.2 / 33.9 / **49.2** |
| | Inception V3 | 56.0 / 50.3 / **91.6** | 35.9 / 32.0 / **69.5** | 44.4 / 35.1 / **84.9** | 45.9 / 35.4 / **77.4** | 65.1 / 57.7 / **75.6** | 49.5 / 42.1 / **79.8** |
| | DenseNet121 | 42.8 / 42.3 / **64.0** | 26.4 / 25.2 / **44.2** | 28.0 / 28.8 / **34.0** | 41.9 / 48.2 / **76.0** | 54.2 / 48.8 / **60.2** | 38.7 / 38.7 / **55.7** |
| | SqueezeNet | 51.7 / 51.1 / **81.1** | 27.9 / 31.6 / **52.5** | 30.2 / 33.1 / **47.1** | 31.6 / 35.1 / **64.2** | 81.3 / 78.9 / **96.4** | 44.5 / 46.0 / **68.3** |
| | Average | 53.8 / 54.8 / **77.2** | 28.1 / 27.4 / **54.2** | 31.3 / 30.1 / **45.0** | 34.6 / 34.7 / **61.9** | 58.3 / 56.1 / **71.4** | 41.2 / 40.6 / **62.0** |

Table 2: **Transferability in the strict black-box setting with no access to target data.** We set $\epsilon = 10$ and report the fooling rate for 5K samples from the ImageNet validation set.

different domain. Furthermore, the resulting generator generalizes particularly well to the unseen target domain when trained with our feature separation loss.

| Discriminator (Model) | Gen. Training (data) | VGG16 | ResNet152 | Inception-v3 | DenseNet121 | SqueezeNet1.1 | Average |
|---|---|---|---|---|---|---|---|
| | | GAP [10] / CDA [11] / Ours | | | | | |
| ChestXNet ( trained on ChestX) | Comics40K | 65.4 / 58.2 / **77.3** | 50.5 / 38.9 / **57.4** | 55.4 / 46.5 / **60.2** | 61.4 / 48.1 / **76.9** | 71.6 / 68.0 / **79.7** | 60.9 / 51.9 / **70.3** |
| | Comics10K | 66.8 / 58.1 / **77.0** | 50.3 / 38.8 / **55.2** | 55.0 / 45.7 / **59.1** | 62.2 / 49.1 / **74.9** | 70.1 / 68.8 / **78.9** | 60.9 / 52.1 / **69.0** |
| | Paintings80K | 76.6 / 63.2 / **78.2** | 53.8 / 46.2 / **56.8** | 56.8 / 52.9 / **60.2** | 73.7 / 55.3 / **77.7** | 79.3 / 70.2 / **80.5** | 68.0 / 57.5 / **70.7** |
| | Paintings10K | 76.7 / 61.3 / **78.4** | 52.9 / 41.7 / **59.9** | 57.4 / 49.5 / **61.7** | 74.6 / 50.5 / **79.1** | 78.6 / 67.4 / **81.1** | 68.0 / 54.1 / **71.9** |
| | ChestX10k | 43.3 / 48.2 / **66.3** | 29.2 / 23.7 / **49.5** | 24.9 / 26.5 / **48.5** | 35.3 / 28.8 / **65.0** | 52.8 / 45.5 / **72.5** | 37.1 / 34.5 / **60.4** |
| | Average | 65.7 / 57.8 / **75.4** | 47.3 / 37.8 / **55.8** | 49.9 / 44.2 / **57.8** | 61.4 / 46.4 / **74.7** | 70.5 / 64.0 / **78.5** | 59.0 / 50.0 / **68.5** |

Table 3: **Extreme cross-domain transferability analysis using a generator trained with ChestXNet and different datasets.** We set $\epsilon = 10$ and report the fooling rates on 5K ImageNet val-set samples.

## 4.3 Extreme Cross-Domain Transferability

We further study the transferability of adversarial attacks in the extreme scenario where neither the target architecture nor the target data are available, and where even the classifier used to learn the generator was trained on data that differs from the target domains. To this end, we use ChestXNet [29] pretrained on ChestX dataset as classifier. We train the generator with different datasets on dense-block12 and evaluate its transferability in Table 3. Despite the challenging nature of this setting, our method still yields satisfying fooling rate; 68.5pp on average vs 87.0pp in the standard black-box scenario. Furthermore, our approach outperforms the state-of-the-art CDA by 18.5% on average.

## 4.4 Transferability to Robust Models

In this section, we study the transferability of our generators to five state-of-the-art defenses, i.e., the high-level representation guided denoiser (HGD) [36], the input preprocessing defense through random resizing and padding (R&P) [37], Feature Denoising (FD) on ResNeXt-101 [38], Projected Gradient Descent (PGD) [39] on ResNet50, and the average of three ensembles of adversarially trained Inception models [40] (EnsembleAdv). In addition to GAP and CDA, we compare our method to RHP, which constitute the state-of-the-art universal attack in terms of effectiveness against defenses. As shown in Table 4, Ours achieves the overall best performance on HGD, R&P and EnsembleAdv. However, we observe that none of the methods are successful in attacking Feature denoising and PGD defenses. Note, however, that with PGD and FD, the error rate on the clean samples is significantly higher than with other defenses, making these strategies ill-suited for practical applications.

| Nework | Method | EnsembleAdv [40] | HGD [36] | RP [37] | FD [38] | PGD [39] |
|---|---|---|---|---|---|---|
| Error on normal samples | | 24.6 | 19.5 | 21.0 | 51.1 | 53.7 |
| Inc-v3 | RHP | 29.5 | 26.8 | 23.3 | 2.38 | 2.40 |
| Inc-v4 | RHP | 25.1 | 23.4 | 20.2 | 1.90 | 2.20 |
| IncRes-v2 | RHP | 28.8 | 26.9 | 25.1 | 2.20 | 2.20 |
| SqueezeNet | GAP | 27.2 | 33.1 | 26.7 | 4.24 | **5.84** |
| SqueezeNet | CDA | 25.6 | 31.4 | 26.5 | 4.06 | 5.42 |
| SqueezeNet | Ours | **33.7** | **43.9** | **32.7** | **7.06** | 3.30 |

Table 4: **Transferability to adversarially trained models**. We report the absolute percentage increase in top-1 error with $\epsilon = 16$ on the 5K samples from ImageNet val-set following [8].

| Gen. Training (data) | Discriminator (Trained on ImageNet) | VGG16 | ResNet50 | EfficientNet | MobileNet-v3 | Average |
|---|---|---|---|---|---|---|
| | | GAP [10] / CDA [11] / Ours | | | | |
| - | No Attack | 68.12 | 66.08 | 61.07 | 55.44 | 62.68 |
| Comics (40K) | VGG-16 | 14.9 / 22.8 / **3.60** | 14.2 / 20.7 / **7.55** | 11.3 / 12.6 / **8.00** | 07.5 / 11.4 / **4.30** | 12.0 / 16.9 / **5.86** |
| | ResNet152 | 32.1 / 24.0 / **9.08** | 25.5 / 16.9 / **8.13** | 30.6 / 15.7 / **7.66** | 21.4 / 13.0 / **5.44** | 27.4 / 17.4 / **7.58** |
| | Inception-v3 | 33.7 / 32.7 / **18.0** | 29.0 / 28.5 / **20.2** | 32.3 / 31.3 / **14.5** | 22.7 / 22.1 / **14.0** | 29.4 / 28.6 / **16.6** |
| | DenseNet121 | 25.1 / 23.2 / **8.08** | 21.8 / 19.8 / **9.28** | 22.4 / 20.7 / **12.9** | 16.8 / 15.3 / **5.60** | 21.5 / 19.8 / **8.96** |
| | SqueezeNet | 28.5 / 26.8 / **13.7** | 23.1 / 24.4 / **11.3** | 17.8 / 20.8 / **12.8** | 14.2 / 18.3 / **6.53** | 20.9 / 22.6 / **11.1** |
| | Average | 26.8 / 25.9 / **10.5** | 22.7 / 22.1 / **11.3** | 22.9 / 20.2 / **11.2** | 16.5 / 16.0 / **7.17** | 22.2 / 21.1 / **10.0** |
| Paintings (80K) | VGG-16 | 16.8 / 16.0 / **8.05** | 18.2 / 17.4 / **12.1** | 11.6 / **10.3** / 11.6 | **7.86** / 7.99 / 8.47 | 13.6 / 12.9 / **10.1** |
| | ResNet152 | 31.0 / 19.6 / **9.66** | 20.5 / 17.1 / **9.31** | 17.8 / 12.2 / **7.74** | 15.1 / 12.1 / **5.45** | 21.1 / 15.3 / **8.04** |
| | Inception-v3 | 32.9 / 33.0 / **16.0** | 28.4 / 28.7 / **18.9** | 27.4 / 27.9 / **14.2** | 22.6 / 21.4 / **12.1** | 27.8 / 27.8 / **15.3** |
| | DenseNet121 | 28.5 / 29.1 / **8.14** | 20.2 / 20.6 / **8.99** | 19.3 / 20.2 / **7.90** | 15.4 / 16.0 / **5.98** | 20.8 / 21.5 / **7.75** |
| | SqueezeNet | 29.6 / 29.1 / **13.6** | 23.8 / 23.8 / **11.4** | 21.0 / 19.4 / **12.9** | 15.9 / 15.0 / **7.69** | 22.6 / 21.8 / **11.4** |
| | Average | 27.7 / 25.4 / **11.1** | 22.2 / 21.5 / **12.1** | 19.4 / 18.0 / **10.9** | 15.4 / 14.5 / **7.94** | 21.2 / 19.8 / **10.5** |
| ImageNet (1.2M) | VGG-16 | 15.6 / 13.5 / **8.30** | 17.8 / 12.1 / **11.8** | 9.19 / **8.42** / 11.4 | 7.76 / **4.82** / 9.12 | 12.6 / **9.71** / 10.2 |
| | ResNet152 | 17.8 / 16.9 / **8.56** | 13.3 / 13.7 / **7.59** | 11.0 / 9.32 / **6.15** | 12.2 / 6.67 / **3.62** | 13.5 / 11.6 / **6.48** |
| | Inception-v3 | **12.8** / 20.1 / 15.8 | 15.2 / 19.1 / **18.4** | 12.4 / **12.0** / 13.7 | 11.8 / 13.6 / **9.96** | **13.0** / 16.2 / 14.5 |
| | DenseNet121 | 20.3 / 19.9 / **4.73** | 14.0 / 14.6 / **6.37** | 11.7 / 11.6 / **6.01** | 11.4 / 11.2 / **2.86** | 14.4 / 14.3 / **5.02** |
| | SqueezeNet1 | 25.1 / 24.5 / **13.1** | 21.8 / 20.4 / **10.8** | 17.1 / 20.1 / **11.5** | 12.7 / 15.4 / **6.19** | 19.2 / 20.1 / **10.4** |
| | Average | 18.3 / 19.0 / **10.1** | 16.4 / 16.0 / **11.0** | 12.3 / 12.3 / **9.77** | 11.2 / 10.3 / **6.35** | 14.5 / 14.4 / **9.31** |

Table 5: **Transferability across tasks with access to neither the target data nor the target model.** We set $\epsilon = 16$ and report the mAP for 4952 samples from the PASCAL VOC test-set.

## 4.5 Cross-Task Transferability Analysis

To demonstrate the transferability of our approach across tasks, we attack object detectors using perturbation generators trained with image classifiers. Specifically, we choose the SSD framework with 4 different backbones, namely, VGG16, ResNet50, EfficientNet, and MobileNet-v3 pretrained on PASCAL VOC [41]. In Table 5, we report the mAP on the PASCAL VOC test-set, containing 4952 images, for perturbation generators trained with different image classifiers and with $\epsilon = 16$. The performance of the SSD detector on clean images is 68.1, 66.1, 61.1 and 55.4 using VGG16, ResNet50, EfficientNet and MobileNetv3 as backbone, respectively. Our attacks significantly decrease these score and yield lower mAPs than the baselines in most cases. These results highlight, for example, that a generator trained on the synthetic Comics dataset with an image classifier can fool an SSD detector trained on a different domain, with a different architecture, and for a different task. As evidenced by the comparisons with CDA and GAP, learning to generate perturbations that affect the mid-level features is more effective than focusing on the classification boundary.

## 4.6 Additional Analysis

To further analyze our method, we visualize the resulting perturbed images using t-SNE plots. Specifically, in Figure 7(a), we provide t-SNE plots obtained using the final latent representation of 1000 normal (blue) and attacked (yellow) ImageNet images for the strict black-box and extreme black-box scenarios. In both cases, our method yields features that clearly separate the normal and adversarial images with large margins, whereas CDA fails to do so. In Figure 8, we visualize the unbounded adversarial images obtained with different methods and also show the bounded adversarial images computed with $\epsilon = 10$ with our approach in the last column. As can also be seen in Figure 9, the perturbations follow repeating patterns corresponding to the patterns of the most disrupted mid-level filters. Note that RHP [8] explicitly enforces perturbation homogeneity using a predefined pattern, such as vertical or horizontal patterns. Our filter visualizations for multiple networks reveals that the bottom layers learn such horizontal and vertical stripes. While we tried training the generator to perturb such layers, we observed that larger $\epsilon$ strength are required in this case. Overall, our results evidence that attacking mid-level filters that have strong correlations across architectures consistently improves the transfer rates.

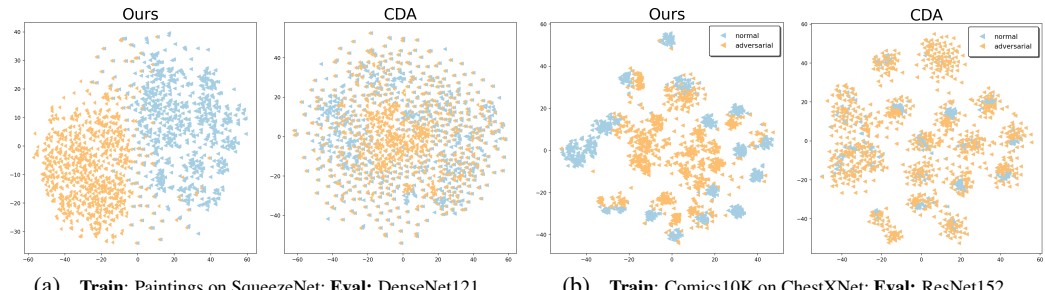

(a) **Train**: Paintings on SqueezeNet; **Eval:** DenseNet121        (b) **Train**: Comics10K on ChestXNet; **Eval:** ResNet152

Figure 7: **t-SNE visualizations.** We show t-SNE plots for 1000 normal (blue) images and their adversarial (yellow) counterparts in **(a)** the strict black-box setting and **(b)** the extreme black-box setting. More visualizations can be found in the supplementary material.

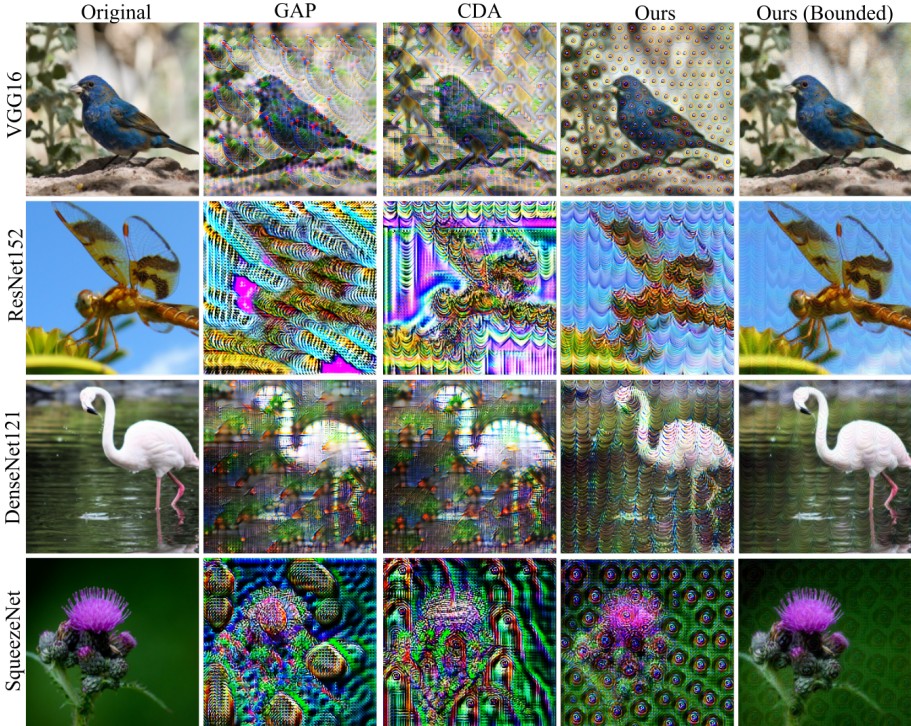

Figure 8: **Qualitative Results.** Visualization of the unbounded outputs obtained with generators trained on ImageNet with different methods. Our approach produces images that are structured and better retain the content of the original images than the baselines. Best viewed in color and zoomed.

**Limitations.** In our experiments, we choose a single layer $l$ at a time. To this end, we sweep over each block of layers, typically around 5 on average for the studied architectures. We empirically found that the best-performing layer is independent of the choice of the target model to attack. For example, a generator trained with a feature separation loss on the conv4-1 layer in VGG16 transfers to all attack settings at inference time, avoiding repeated sweeps and computational costs. For all our experiments, we set the layer $l$ to attack to relu after conv4-1, layer3, mixed6b, denseblock8, and fire10 for VGG16, ResNet152, Inception-v3, DenseNet121, and SqueezeNet, respectively. We report additional results with generators trained at different layers in the supplementary material. Nevertheless, we have no guarantees that such layers truly are optimal; in particular, better results might be achievable by attacking ensembles of layers, but this would require tuning the layer combination.

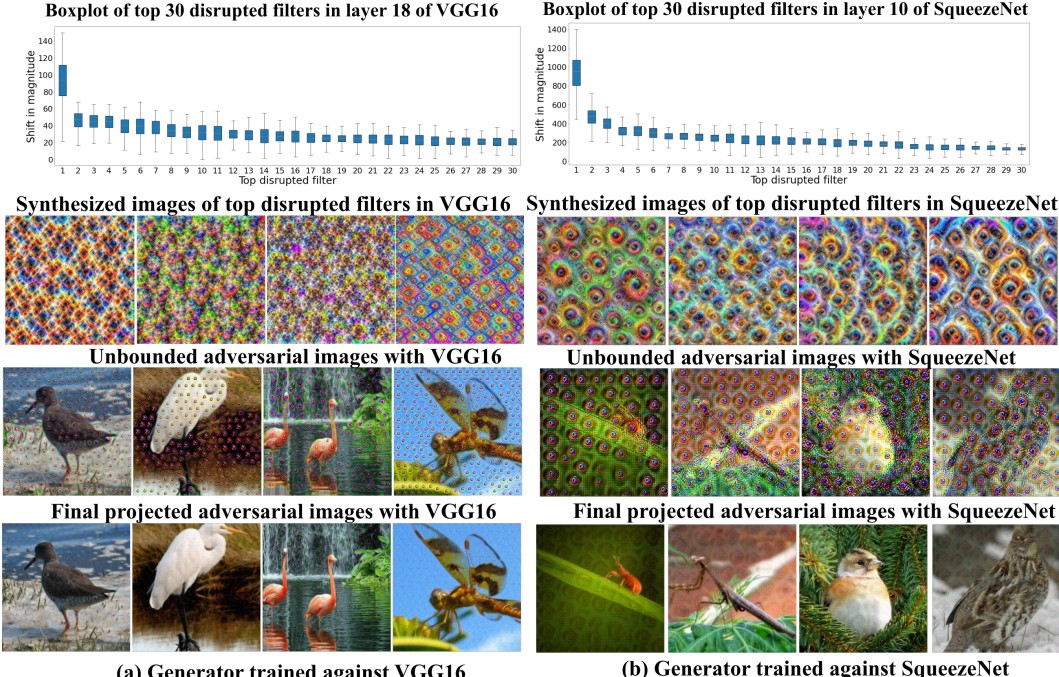

**(a) Generator trained against VGG16**

**(b) Generator trained against SqueezeNet**

Figure 9: **Correlation between adversarial patterns and most disrupted filters.** The first row shows box plots for the shift in magnitude of the top 30 disrupted filters, i.e., with the largest shift. In the second row, we show the corresponding visualizations obtained using [12] for the top 4 filters. The third row evidences the strong correlation between the top-disrupted filters and the unbounded adversarial images. In the last row, we visualize the bounded adversaries for $\epsilon = 10$. Comparing the left and right blocks shows that the disrupted filters in VGG16 and SqueezeNet contain visually-similar patterns, thereby explaining our high transfer rates. Best viewed in color and zoomed.

# 5 Conclusion

In this paper, we have explored the use of mid-level features in conjunction with generative networks to learn transferable perturbations. We have shown that a generator trained with feature separation loss can successfully fool models across architectures and tasks, and that a proxy dataset from a different domain can be leveraged to learn effective perturbations. Our experiments demonstrate that our method outperforms the state-of-the-art attacks across a wide range of architectures and tasks. We have limited our experiments to undefended models due to the unavailability of publicly available robust models across architectures but will study this in future work. Furthermore, we believe that a deeper analysis of learned filter banks with respect to changes in architectures can shed light on building better black-box models.

# 6 Broader Impact

Understanding the strength of adversarial attacks is critical to building robust defenses in the future. The practical implications of adversarial examples in safety-critical applications, such as autonomous driving and biometrics, are huge. Our extensive set of experiments highlights the vulnerability of modern networks to adversarial attacks even in the extreme black-box case, which has a high potential for societal impact. We hope that our work will motivate the community to further study the internal workings of neural networks in the context of adversarial attacks, and in particular exploit such analysis to design effective defense mechanisms.

## Acknowledgement

This work was supported by the Swiss National Science Foundation.

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
