# Learning Transferable Adversarial Perturbations

## Supplementary Material

**Krishna Kanth Nakka[1], Mathieu Salzmann[1,2]**
[1]CVLab, EPFL, Switzerland
[2]ClearSpace, Switzerland
{krishna.nakka, mathieu.salzmann}@epfl.ch

In this document, we provide further insights and ablation studies of our approach.

**CKA Metric.** In addition to t-SNE visualizations and top activated channels of internal layers in the main paper, we present here in Table 1 the centered kernel alignment (CKA) [1] values to provide additional insights about the similarities between the internal representations of different networks. We observed that the CKA values are typically above 0.75 for the intermediate layers that we considered in our experiments, further explaining the reason for high transferability. For example, on transfer attacks between VGG16 and DenseNet121, the CKA values are highest between the intermediate layer 8 in DenseNet121 and layers 18 and 23 in VGG16. Similarly, between ResNet152 and DenseNet121, the CKA value is 0.76 for the optimal generator configuration, which relies on layer 3 in ResNet152 and layer 8 in DenseNet121.

| Network & | DenseNet121 | | | | | | Network & | DenseNet121 | | | | |
|---|---|---|---|---|---|---|---|---|---|---|---|---|
| Layer | 3 | 6 | 8 | 10 | 12 | | Layer | 3 | 6 | 8 | 10 | 12 |
| VGG16   4 | 0.71 | 0.64 | 0.26 | 0.21 | 0.14 | | ResNet152   0 | 0.69 | 0.54 | 0.25 | 0.20 | 0.13 |
| 9 | 0.48 | 0.58 | 0.42 | 0.30 | 0.19 | | 1 | 0.74 | 0.82 | 0.42 | 0.33 | 0.21 |
| 18 | 0.31 | 0.58 | 0.75 | 0.58 | 0.34 | | 2 | 0.57 | 0.83 | 0.65 | 0.49 | 0.31 |
| 23 | 0.26 | 0.50 | 0.79 | 0.69 | 0.41 | | 3 | 0.24 | 0.45 | 0.76 | 0.80 | 0.56 |
| 30 | 0.11 | 0.23 | 0.50 | 0.58 | 0.45 | | 4 | 0.13 | 0.25 | 0.45 | 0.61 | 0.74 |

Table 1: CKA [1] scores to understand the similarities between the internal representations of different networks sampled at 5 layers for 500 images on ImageNet val-set.

**Iterative versus Generative Attacks.** In Table 2, we compare the effectiveness of our feature separation loss when used in either an iterative approach, i.e., PGD [2], or with our proposed generator-based one. We set $\epsilon = 10$ with a step size 2 and perform 10 iterations for PGD. We observe that the generator-based method significantly outperforms the iterative method by a large margin of 45pp on average.

| Gen. Training (data) | Discriminator (Model) | VGG16 | ResNet152 | Inception-v3 | DenseNet121 | SqueezeNet | Average |
|---|---|---|---|---|---|---|---|
| | | Iterative / Generative | | | | | |
| **ImageNet** | VGG16 | 98.4 / **99.3** | 31.3 / **68.4** | 30.6 / **46.6** | 40.2 / **84.7** | 64.6 / **86.5** | 41.7 / **71.6** |
| | ResNet152 | 47.4 / **99.1** | 99.4 / **99.7** | 30.9 / **74.9** | 43.6 / **98.8** | 51.6 / **89.1** | 43.4 / **90.5** |
| | Inception-v3 | 42.6 / **98.7** | 24.2 / **90.2** | 94.1 / **99.5** | 30.2 / **96.0** | 53.9 / **91.5** | 37.7 / **94.1** |
| | DenseNet121 | 71.2 / **98.4** | 49.1 / **99.7** | 39.2 / **86.0** | 99.6 / **99.6** | 63.6 / **95.6** | 55.7 / **94.9** |
| | SqueezeNet | 46.7 / **96.1** | 22.7 / **76.4** | 24.4 / **70.7** | 28.4 / **88.7** | 99.7 / **99.7** | 30.5 / **83.0** |
| | Average | | | | | | 41.8 / **86.8** |

Table 2: **Comparison to an iterative approach.** We set $\epsilon = 10$ and report the fooling rates on 5K ImageNet val-set samples using our feature separation loss with either a 10-step PGD attack or our generator-based one trained on ImageNet data. The generator-based method consistently outperforms the iterative one.

**Additional Visualizations.**    Firstly in Figure 1, we show few example images from each domain to understand its characteristics. We observe that ChestX contains larger domain shift than Comics and Paintings to ImageNet. In Figure 2, we show the reasons for high transfer rates between ResNet152 and VGG16 architectures due to similar mid-level filter bank. In Figures 3, 4 and 5, we show the unbounded adversarial images obtained by attacking diffferent layer positions against VGG16, SqueezeNet and DenseNet121, respectively. In addition, we also visualize the unbounded adversarial images in strict black-box setting in Figures 6, 7, 8 and 9 on VGG16, SqueezeNet, ResNet152 and DenseNet121, respectively. In Figure 10, we visualize the output detections of SSD on different backbones with generators trained against DenseNet121 on ImageNet. In Figure 11 and 12, we provide the t-SNE visualizations of the final features in the standard black-box setting with generators trained on Comics and Paintings, respectively. We set the discriminator to SqueezeNet and compare our approach, shown in the first row, with CDA [3] and GAP [4] in the second and third rows. Overall, we observe a clearer separation between the normal and adversarial features using our method than with the baselines.

**Ablation Study.**    Finally, we perform an ablation study of our generator-based approach by attacking the features at every block of layers in all the studied models. For this experiment, we train the generator with 10K samples taken from ImageNet [5], Comics [6] and Paintings [7]. In Figure 13, we plot the fooling rates obtained when training the generator with SqueezeNet [8], VGG16 [9], ResNet152 [10], Inception-v3 [11] and DenseNet [12], each model corresponding to one row of the figure. The columns in each row correspond to the different datasets. In Figure 14, we provide the results of a similar study but in the cross-task setting of attacking the SSD detector with 4 different backbones, and thus report the mAP. Overall, the results indicate a good transferability rate for all target models and datasets. Furthermore, we observe that setting the layer $l$ position to conv4-1 with ReLU (18), layer3 (3), mixed6b (11), denseblock8 (8), and fire10 (10) for VGG16, ResNet152, Inception-v3, DenseNet121, and SqueezeNet, respectively, outperforms the other layers in almost all the cases. The only exception arises in the cross-task scenario when training the generator with a VGG16 on ImageNet10K, in which case the optimal success rate is achieved with layer 23 instead of layer 18. These results further confirm that attacking either the initial layers or the final ones is suboptimal for attack transfer.

**Implementation Details.**    We train the ResNet generator following the same architecture as in CDA [3] containing 6 residual blocks using Adam optimizer with $\beta_1 = 0.5$ and $\beta_2 = 0.99$ with learning rate set to 0.0002 and batch size of 16. For all experiments, we train the generator for 1 epoch (80K iterations) on ImageNet1.2M, 10 epochs on Comics40K, and Paintings80K and 50 epochs on ChestX8K. Further, we decay the learning rate by 0.3 at 30 epoch on training with ChestX. We typically warm-start the generators when trained with ResNet and DenseNet discriminators with lower $\epsilon = 4$ for 2K iterations. For attacking adversarially trained models, we perform gaussian smoothing similar to in CDA [3]. Similar to in [3], during the evaluation of the ImageNet target classifiers, we resize the images to the resolution of the source model before passing them to the generator. For training the SSD models, we set the number of iterations to 120K and use an SGD optimizer with momentum 0.9, weight decay 0.0005, and batch size 32. The learning rate starts at 0.001 and is dropped by a factor of 10 at 80K and 100K iterations. During the evaluation phase, we set the confidence threshold to 0.5 to compute the mAP score.

(a) ChestX

(b) Paintings

(c) Comics

(d) ImageNet

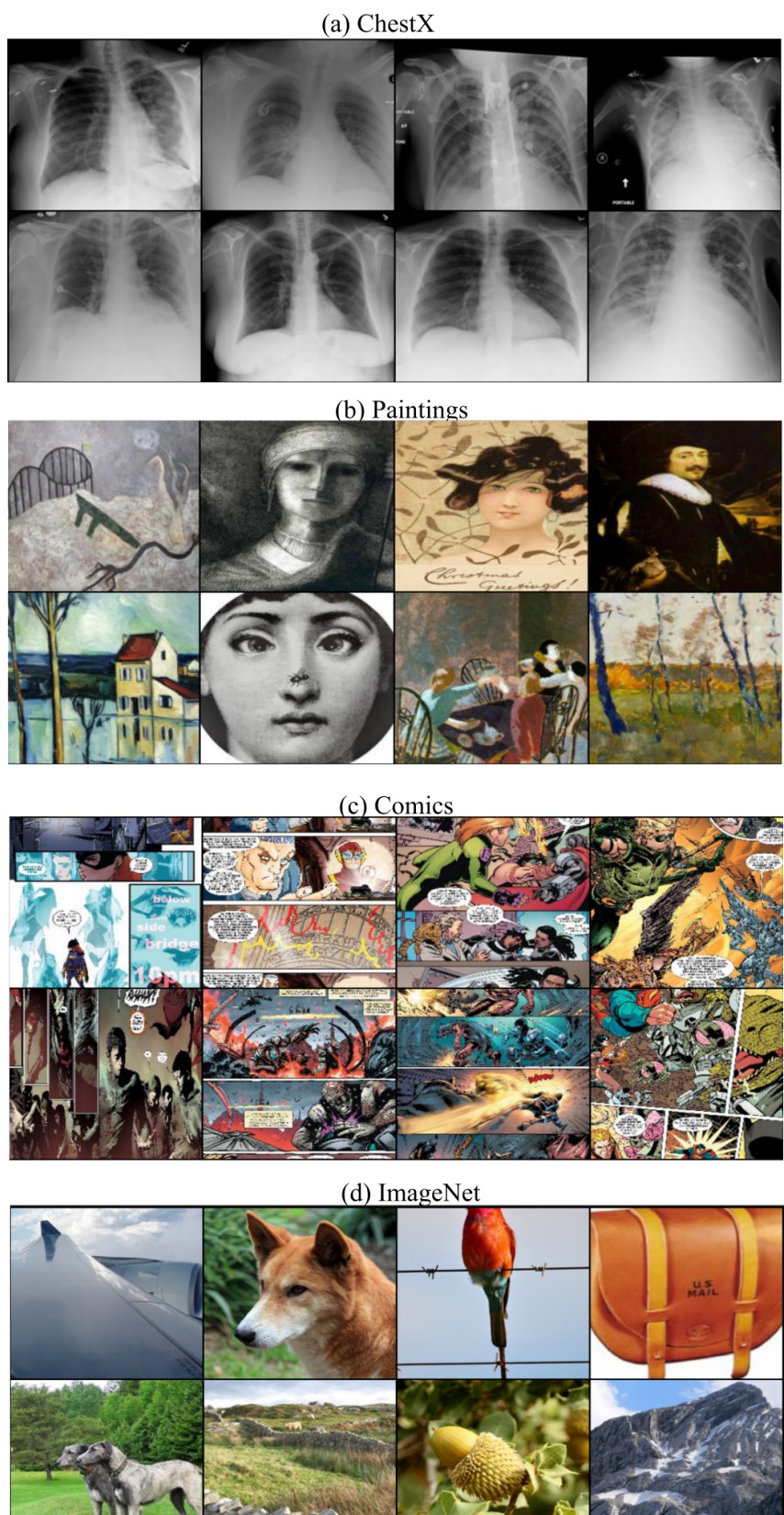

Figure 1: **Sample Images.** Visualization of example images from each domain. ChestX domain focuses on lung region with high domain shift to ImageNet and does not have color. Comics domain contains cartoon images and the Paintings domain mainly focuses on drawings of human subjects. Both Comics and Paintings are synthetic datasets₃

**(a) White-box attack on ResNet152 (Fooling Rate: 99.7%)**

| Original | Unbounded Adv. | Bounded Adv. |
|---|---|---|

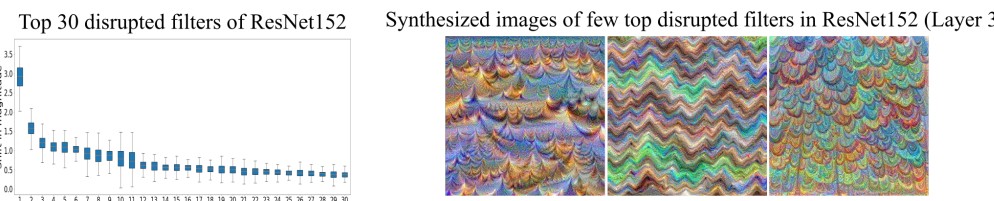

Top 30 disrupted filters of ResNet152     Synthesized images of few top disrupted filters in ResNet152 (Layer 3)

**(b) Transfer  attack from ResNet152  to VGG16 (Fooling Rate: 99.1%)**

Top 30 disrupted filters of VGG16     Synthesized images of few top disrupted filters in VGG16 (Layer 15)

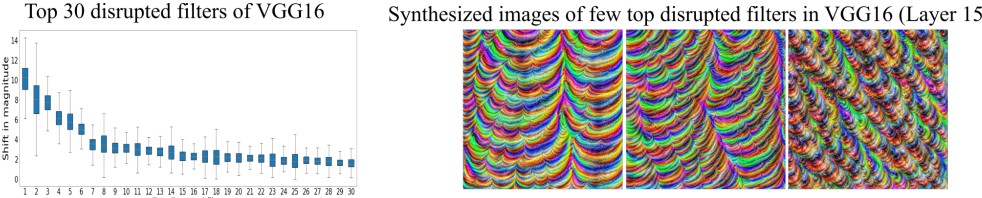

Figure 2: **Black-box transfer from ResNet152 to VGG16.** We analyze the reason for the high transfer rate of 99.1% from ResNet152 to VGG16 by visualizing a few top disrupted filters in intermediate layers. As observed from above, the top disrupted filters in VGG16 and ResNet152 resemble similar texture patterns, and thus disrupting them facilitates high transfer without overfitting to the source model.

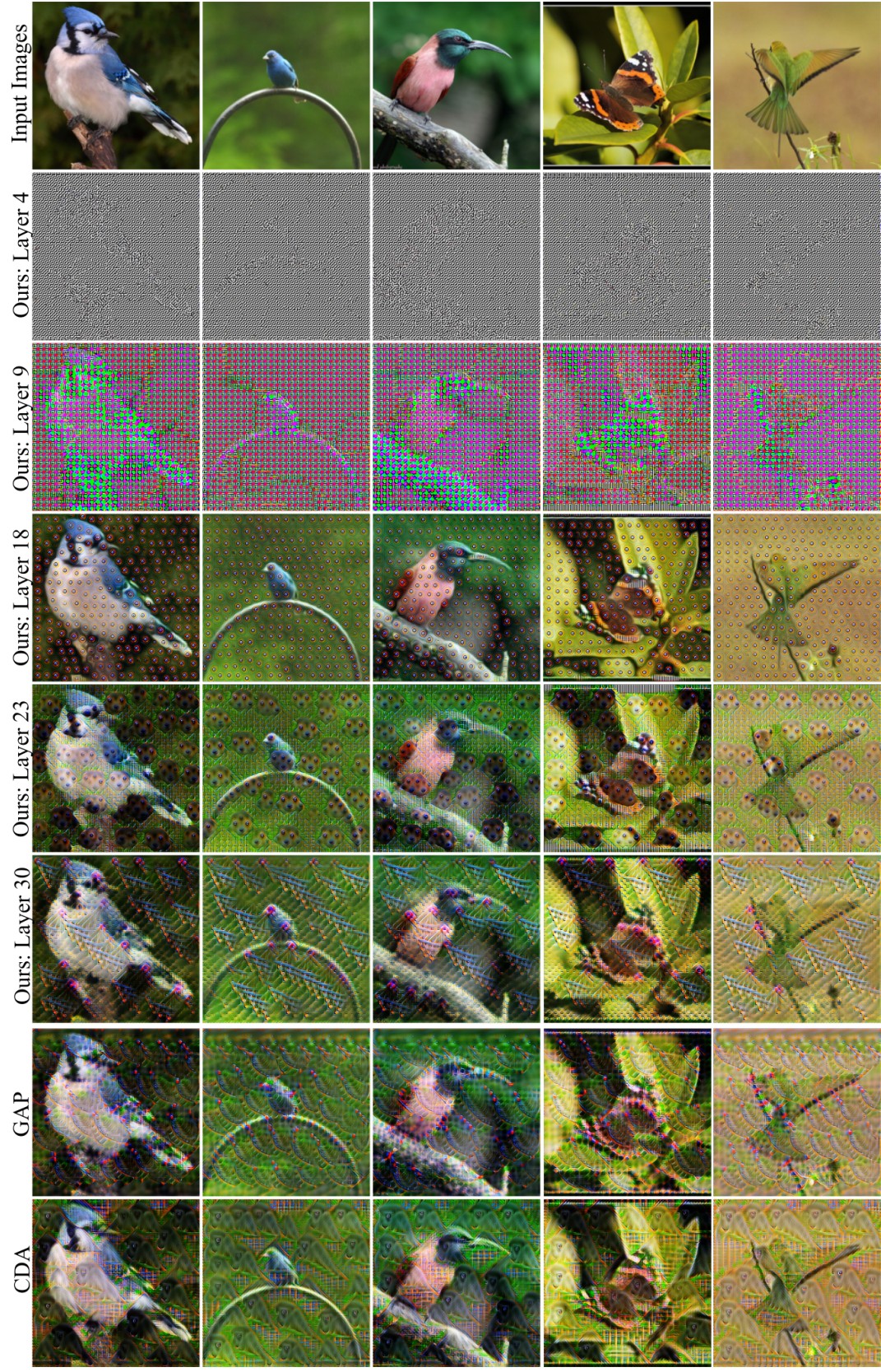

Figure 3: **Qualitative Results by varying the attacked layer position on VGG16.** Visualization of unbounded adversarial outputs at 5 different layers for VGG16 on ImageNet10K training set.

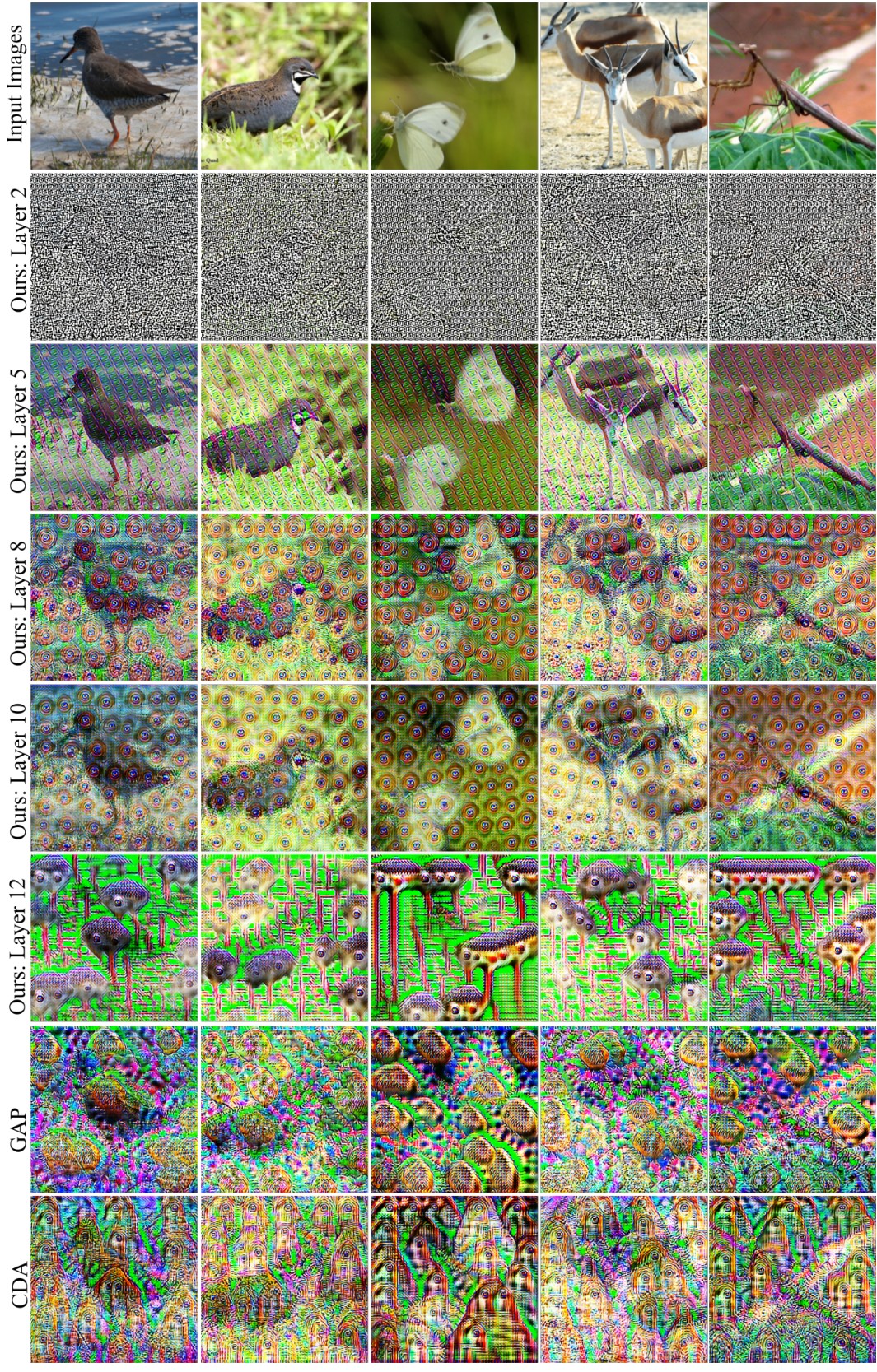

Figure 4: **Qualitative Results by varying the attacked layer position on SqueezeNet1.1.** Visualization of unbounded adversarial ouputs at 5 different layers for SqueezeNet1.1 on ImageNet10K training set.

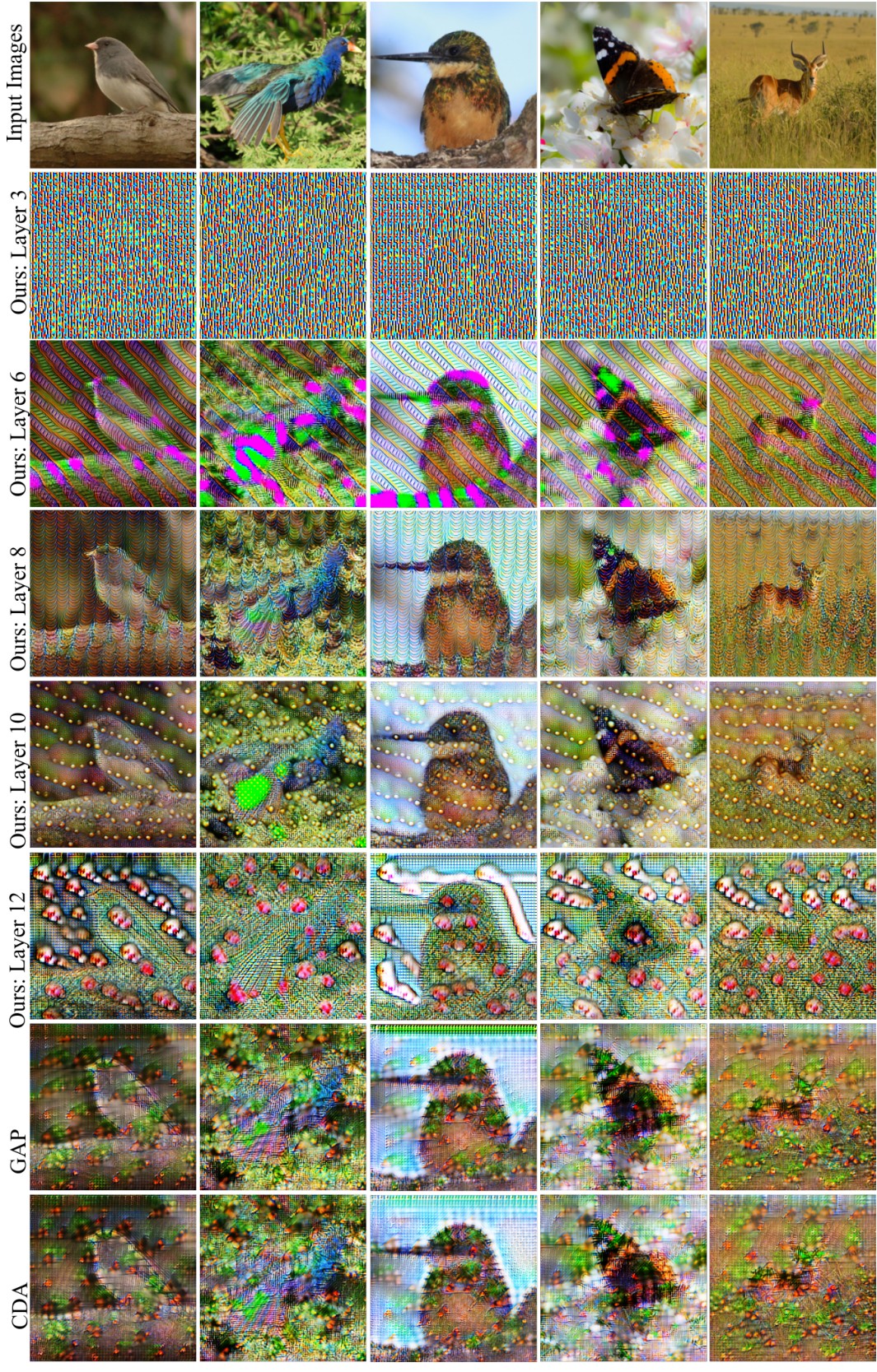

Figure 5: **Qualitative Results by varying the attacked layer position on DenseNet121.** Visualization of unbounded adversarial outputs at 5 different layers for DenseNet121 on ImageNet10K training set.

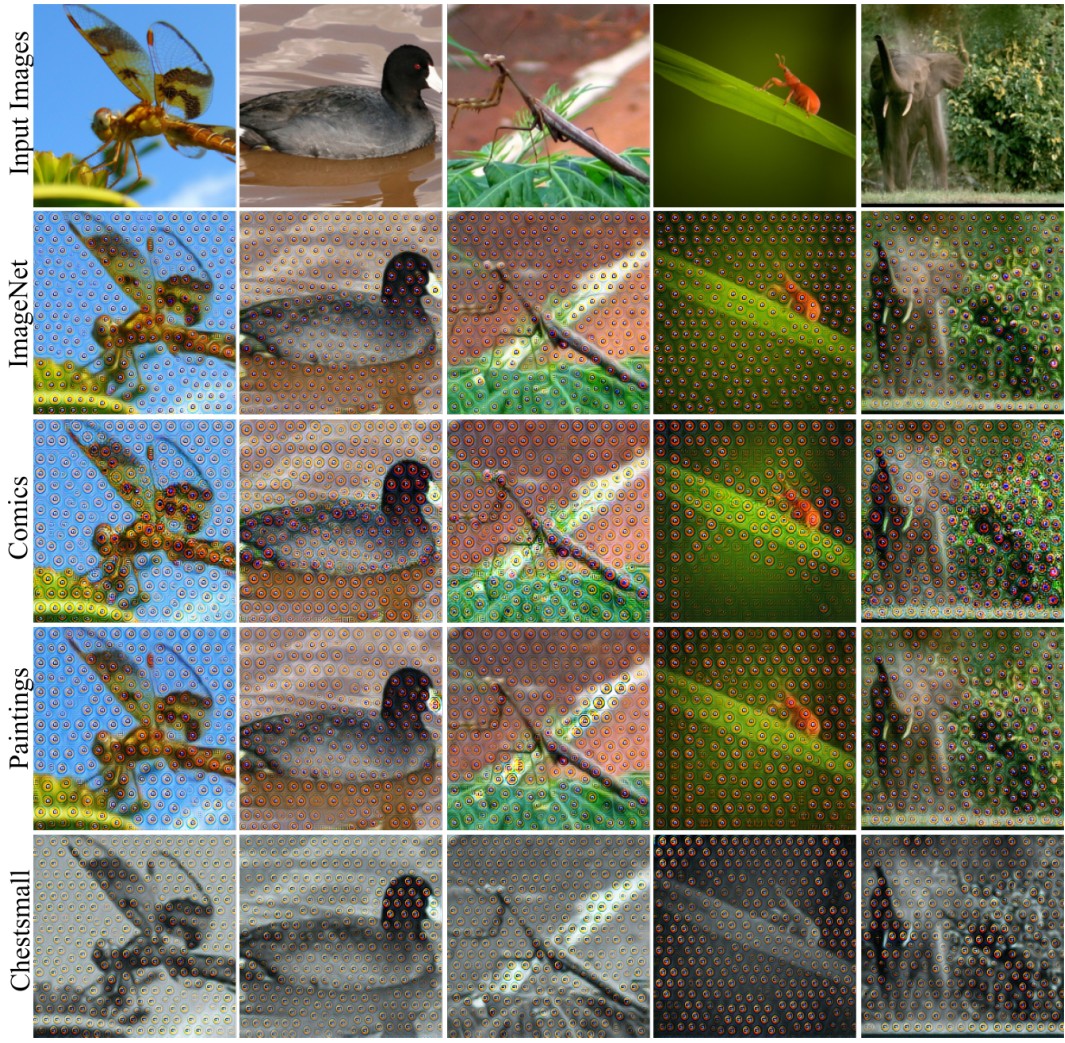

Figure 6: **Qualitative Results by varying the source dataset against VGG16.** Visualization of unbounded adversarial outputs using our approach on VGG16 with different source datasets.

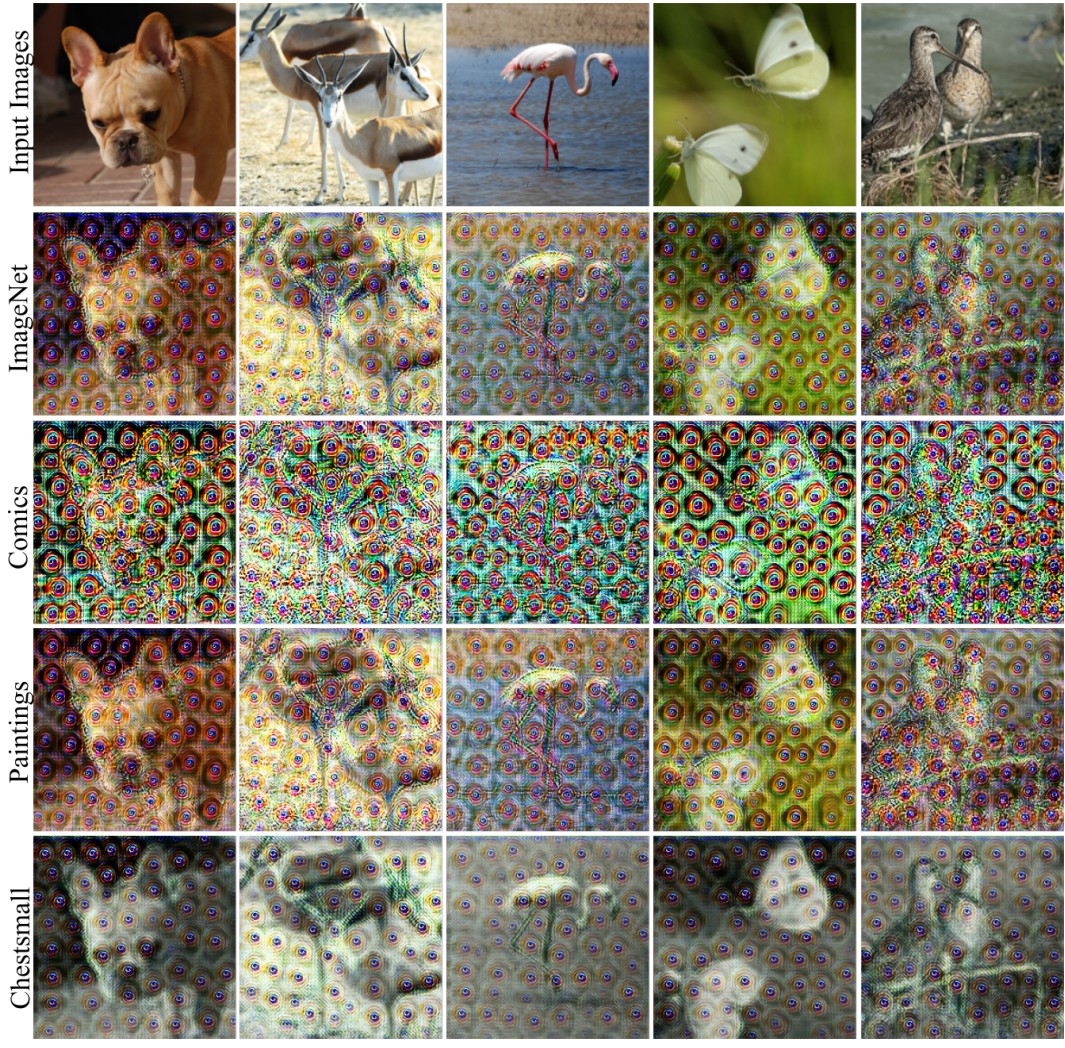

Figure 7: **Qualitative Results by varying the source dataset against SqueezeNet1.1.** Visualization of unbounded adversarial outputs of our approach on SqueezeNet1.1. with different source datasets.

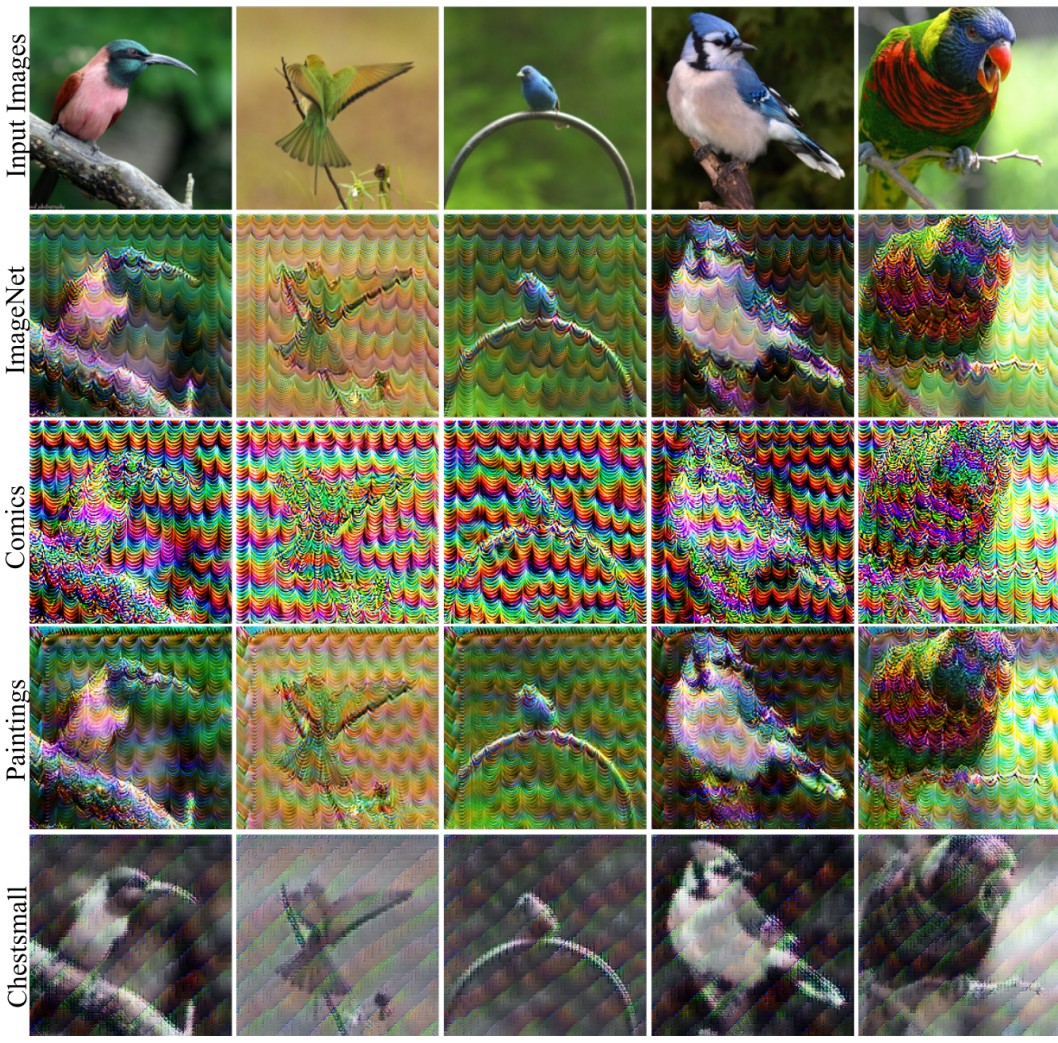

Figure 8: **Qualitative Results by varying the source dataset against ResNet152.** Visualization of unbounded adversarial outputs using our approach for ResNet152 with different source datasets.

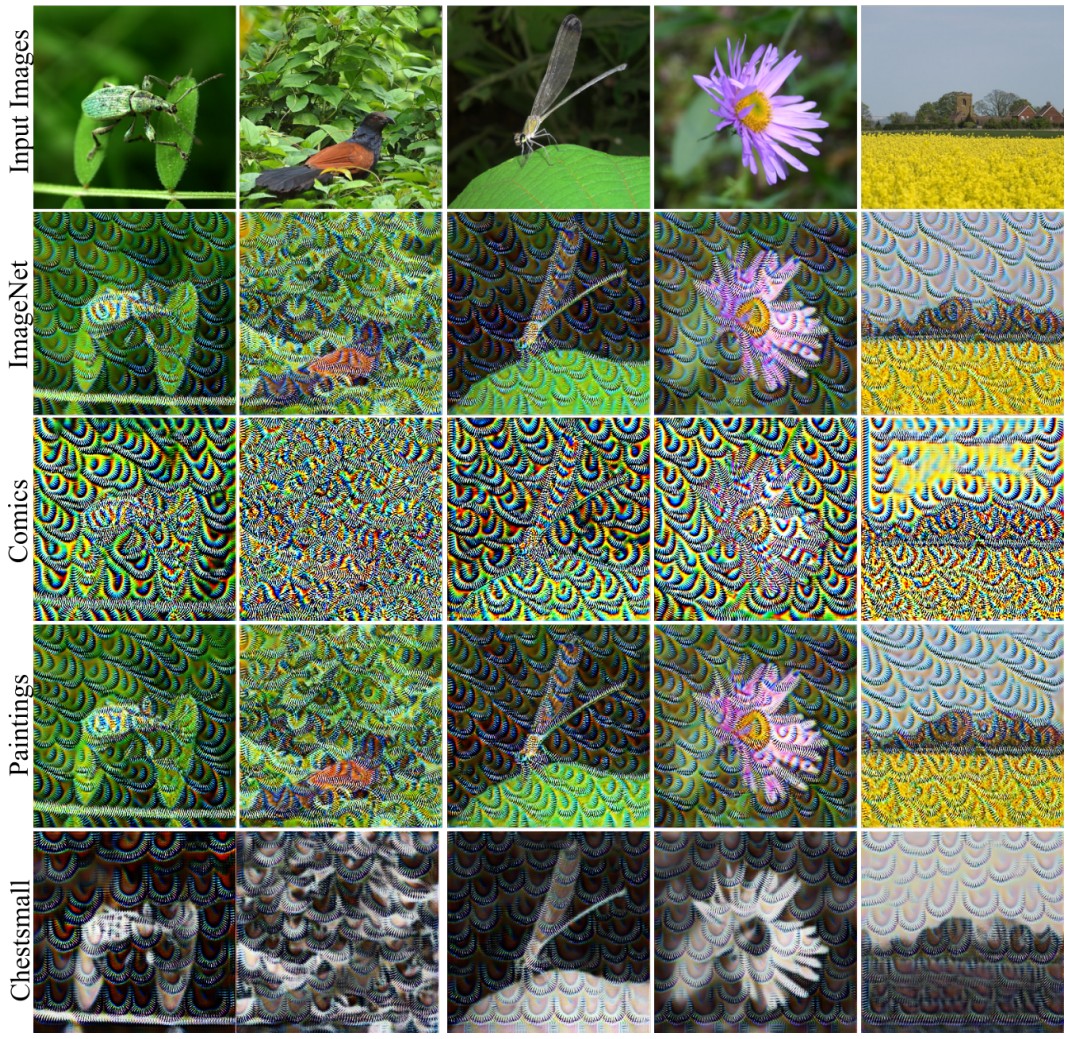

Figure 9: **Qualitative Results by varying the source dataset against Inceptionv3.** Visualization of unbounded adversarial images using our approach for Inceptionv3 with different source datasets.

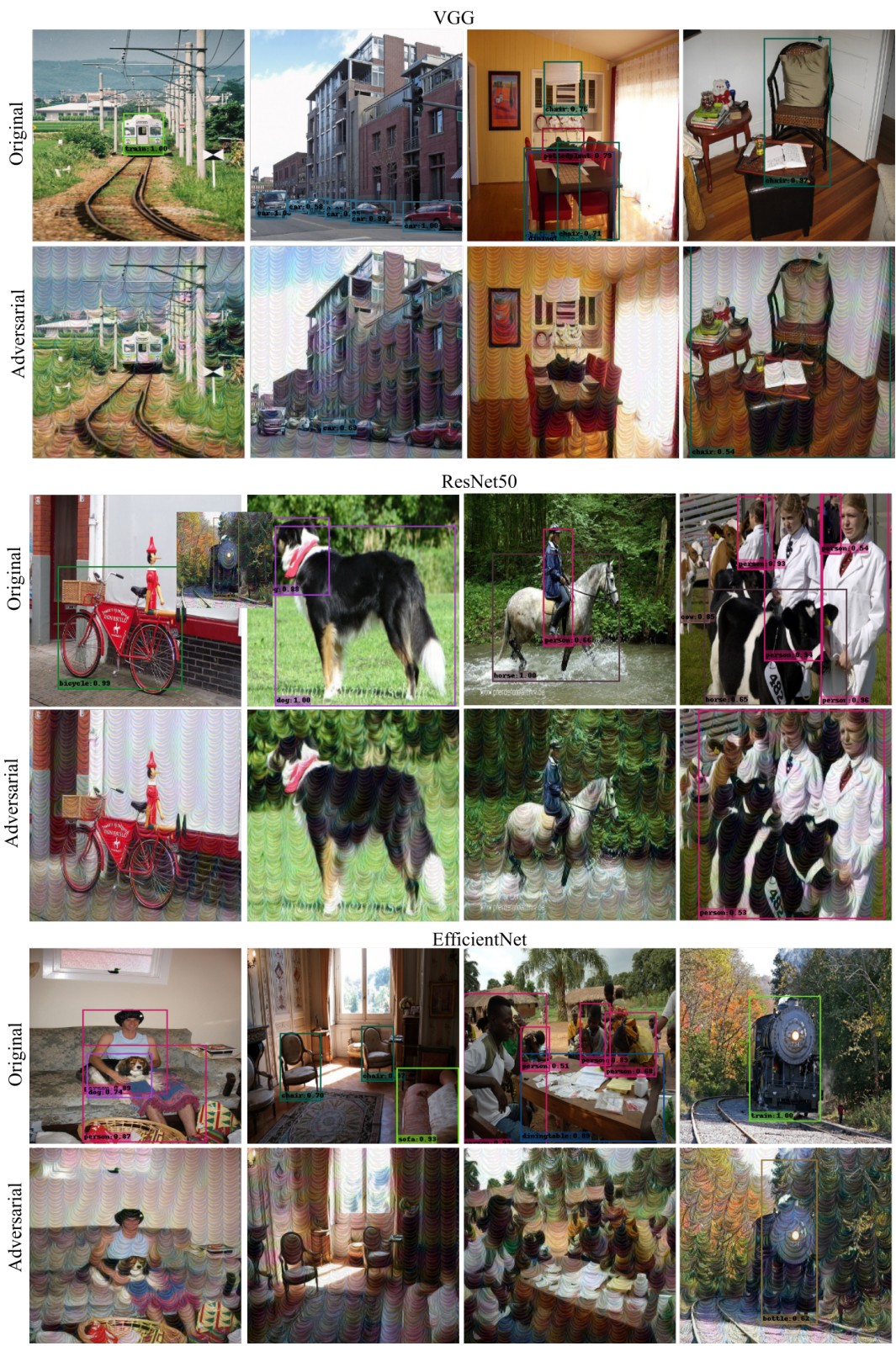

Figure 10: **Qualitative Results on diffferent target task** We visualize the shift in output detections for various backbones on SSD with $\epsilon$ set to 16. The generators are trained with DenseNet121 on ImageNet. Best viewed in color and zoom.

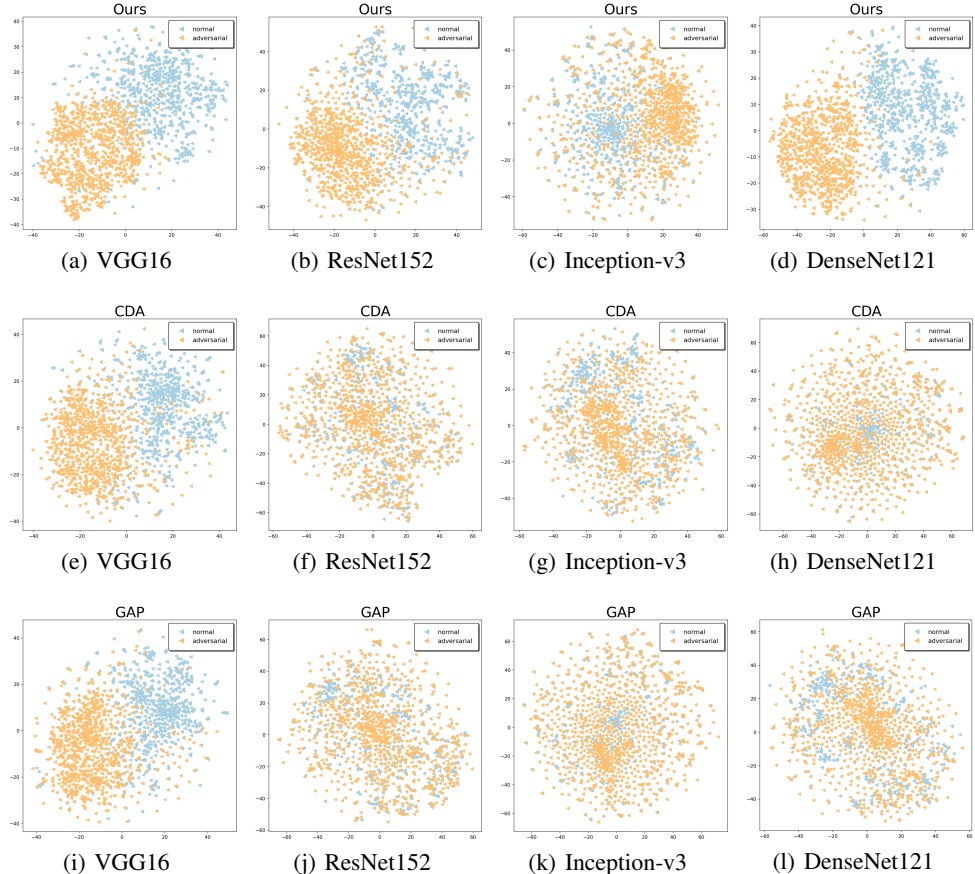

Figure 11: **t-SNE visualizations in the standard black-box setting.** We show t-SNE plots for 1000 normal (blue) ImageNet images and their adversarial (yellow) counterparts in the *standard black-box* setting with a generator trained on *Paintings data with SqueezeNet* discriminator. We provide the visualizations of our method, CDA, and GAP in the first, second, and third row, respectively. Our method separates the features more clearly than the baselines.

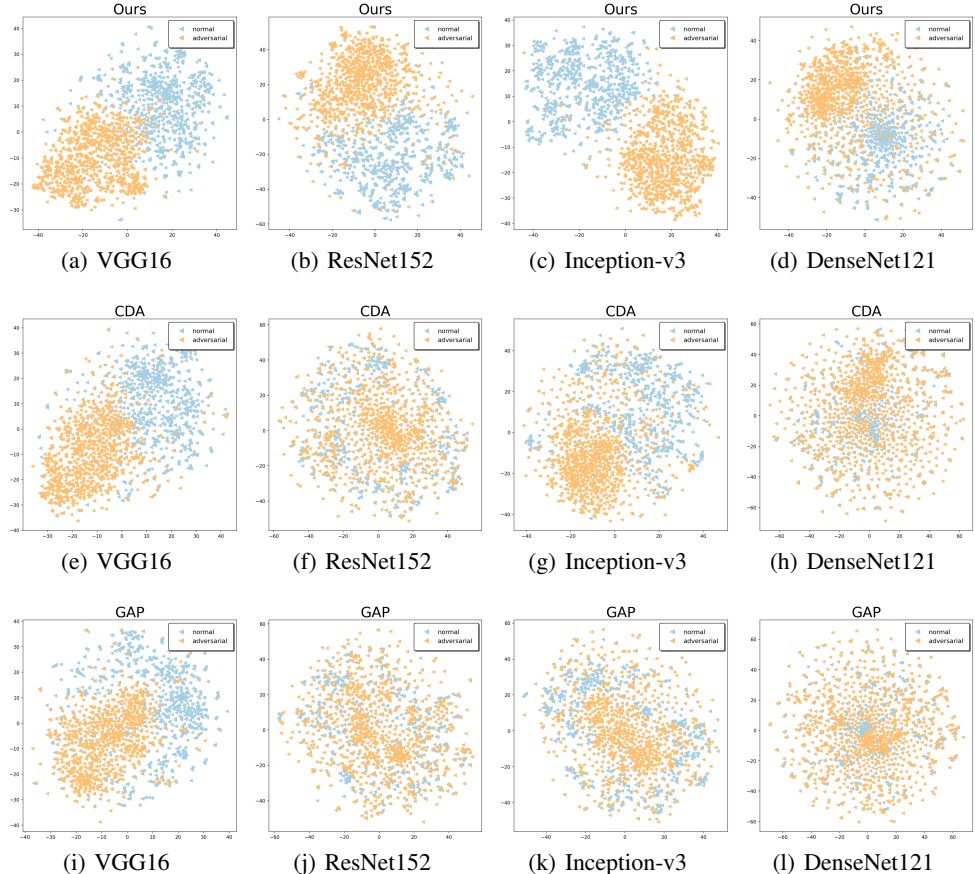

Figure 12: **t-SNE visualizations in the standard black-box setting.** We show t-SNE plots for 1000 normal (blue) ImageNet images and their adversarial (yellow) counterparts in the *standard black-box* setting with a generator trained on *Comics data with SqueezeNet* discriminator. We provide the visualizations of our method, CDA, and GAP in the first, second, and third row, respectively. Our method separates the features more clearly than the baselines.

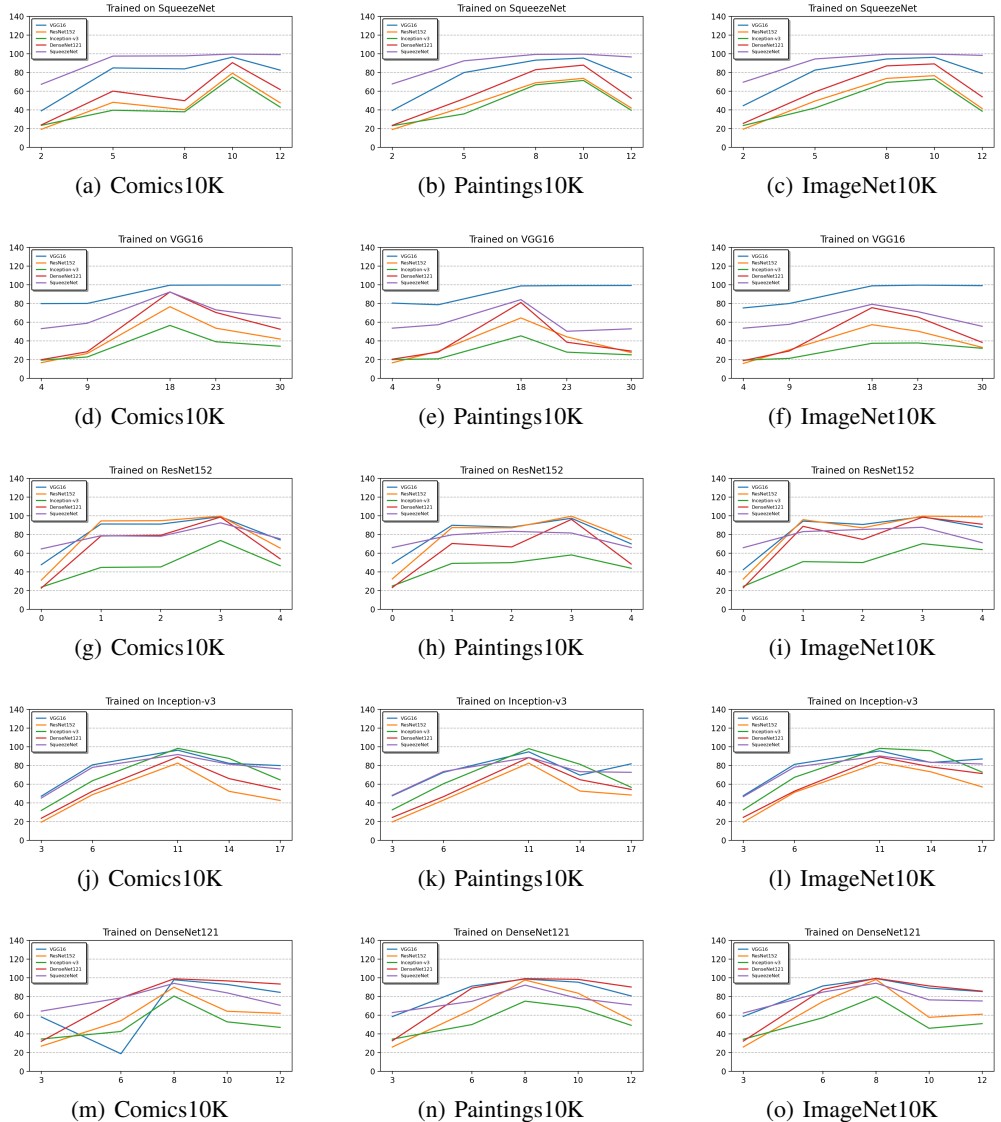

Figure 13: **Impact of the position of the attacked layer in the standard black-box setting.** We vary the position of the layer to attack during training from the bottom layer to the top classification layer. We report, row-wise, the fooling rates (in %) obtained by training the generator with SqueezeNet [8], VGG16 [9], ResNet152 [10], Inception-v3 [11], DenseNet121 [12]. The columns indicate different training sets, containing 10K samples from Comics, Paintings, and ImageNet. In each subplot, we evaluate all 5 networks with $\epsilon = 10$. The best performing layer position is independent of the target architecture and target data. The x-axis represent the layer position and y-axis denote the fooling rate.

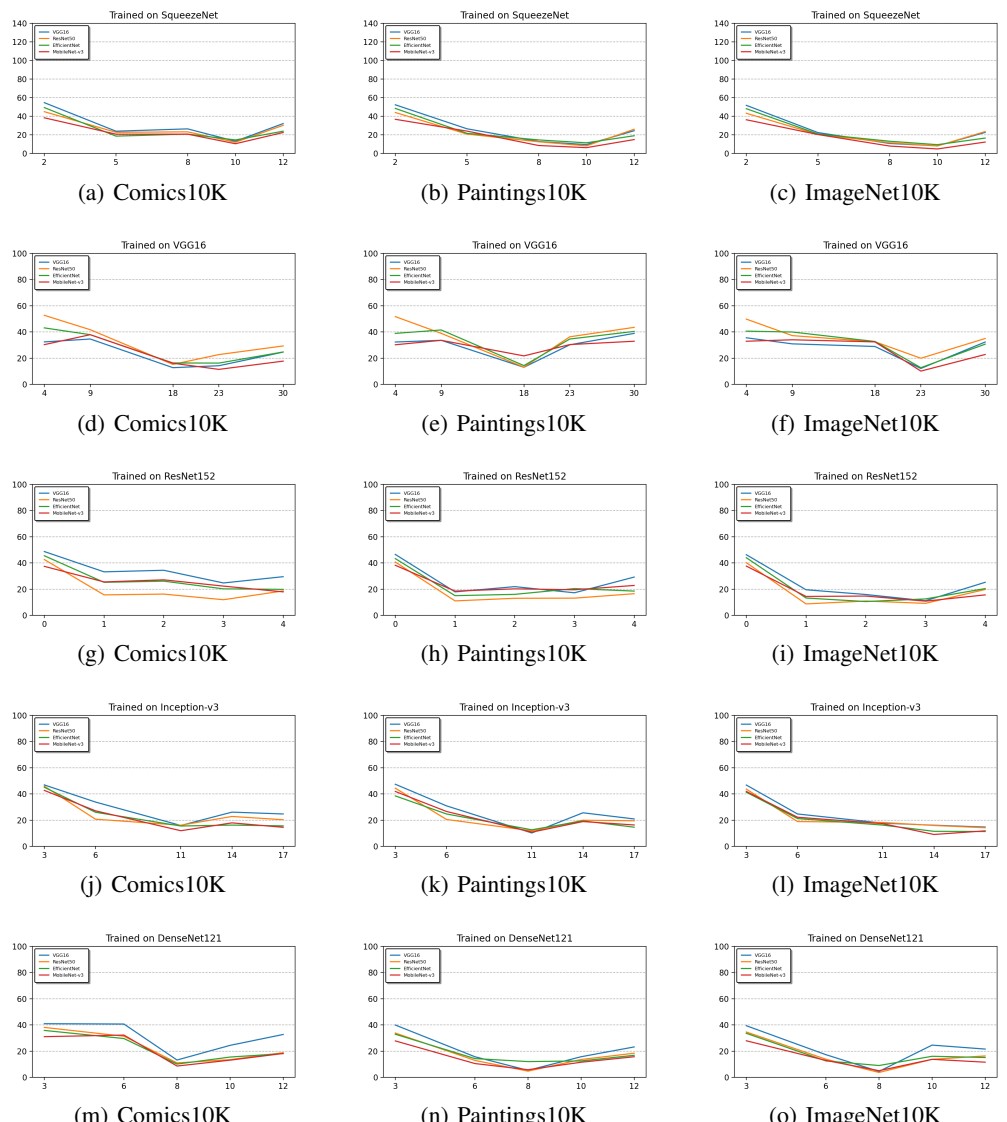

Figure 14: **Impact of the position of the attacked layer in the cross-task setting.** We vary the position of the layer to attack during training from the bottom layer to the top classification layer of the recognition models, and then transfer to evaluate the effectiveness on detection models. We report, row-wise, the mAP obtained by training the generator with SqueezeNet [8], VGG16 [9], ResNet152 [10], Inception-v3 [11], DenseNet121 [12]. The columns indicate different training sets, containing 10K samples from Comics, Paintings, and ImageNet. In each subplot, we evaluate 4 backbones with the SSD framework with $\epsilon = 16$. The best performing layer position is independent of the target task and architecture in 24 out of 25 cases. The x-axis represent the layer position and y-axis denote the mAP metric.