# OpenReview forum: "Learning Transferable Adversarial Perturbations"
_NeurIPS.cc/2021/Conference — NeurIPS 2021 Poster_

### Official Review · Reviewer_WvGo · 2021-07-15

**Rating:** 6
**Confidence:** 4

**Summary:**

This paper shows that mid-level features show a common pattern across different domains/tasks/architectures. Motivated by this, this paper introduced to train a generative model which leveraging the mid-level features to produce adversarial examples. Experimental results demonstrated that the proposed method is effective under different settings.

**Limitations And Societal Impact:**

Authors have adequately addressed the limitations and potential negative societal impact

**Main Review:**

---
Strength:
- The introduced methods show improvements over existing methods (CDA) in several different settings, including a challenging black-box attack setting. The attacker does not know any information about the target architecture/dataset/task.
- The experiments results are well designed, comprehensive and outperform existing methods. Analysis using t-SNE shows the introduced method can produce well-separated features even in cross-domain/architecture attacks.
- The writing and presentations of this paper are easy to follow.
---
Concerns and Questions:
- Although the visualization of mid-level features in Figure 2 shows certain similarities, a qualitative metric would be more convincing. Such as CKA/CCA [1].
- Why not include RHP in Table 1,2,3? It is also a baseline method.
- In Table 3, why is the fooling rate lower for ChestX10k than others? Using ChestX data to train generators against ChestXNet should have a higher fooling rate, and this result is a bit puzzling.
- Figures 3 and 4 in supplementary is the missing label for the x and y-axis. Presumably, x for layers and y for fooling rate, there is no need for >100 for the y axis. Is Figure 3/4 only considered the mid-level layers for different architectures? For ResNet models, layers are chosen from 1-4, while others choose from a wide range of layers. Is there any method/reason for how to choose these layers? The introduced methods rely on the choice of intermediate layers for the attack. There should be a more comprehensive ablation study on this. Such as a simple comparison for introduced method applies to low/high-level features with mid-level features.

[1] Similarity of Neural Network Representations Revisited, ICML 2019.




**Time Spent Reviewing:**

3

---

> ### Author Response · Authors · 2021-08-10
> **Response**
>
> Thank you for your careful reading of our paper and your detailed feedback.
>
> #### **Qualitative Metric**.
>
> Thank you for the suggestion. The CKA values shown below indeed give us additional insights about the similarities between the internal representations of different networks. We observed that the CKA values are typically above 0.75 for the intermediate layers that we considered in our experiments, further explaining the reason for high transferability. For example, on transfer attacks between VGG16 and DenseNet121, the CKA values are highest between the intermediate layer 8 in DenseNet121 and layers 18 and 23 in VGG16. Similarly, between ResNet152 and DenseNet121, the CKA value is 0.76 for the optimal generator configuration, which relies on layer 3 in ResNet152 and layer 8 in DenseNet121. We discuss this in the revised paper.
>
> |           |       |       | DenseNet121 |        |        |               |       |       | DenseNet121 |        |        |
> | :-------: | :---: | :---: | :---------: | :----: | :----: | :-----------: | :---: | :---: | :---------: | :----: | :----: |
> | **VGG16** | **3** | **6** |    **8**    | **10** | **12** | **ResNet152** | **3** | **6** |    **8**    | **10** | **12** |
> |   **4**   | 0.71  | 0.64  |    0.26     |  0.21  |  0.14  |     **0**     | 0.69  | 0.54  |    0.25     |  0.2   |  0.13  |
> |   **9**   | 0.48  | 0.58  |    0.42     |  0.30  |  0.19  |     **1**     | 0.74  | 0.82  |    0.42     |  0.33  |  0.21  |
> |  **18**   | 0.31  | 0.58  |    0.75     |  0.58  |  0.34  |     **2**     | 0.57  | 0.83  |    0.65     |  0.49  |  0.31  |
> |  **23**   | 0.26  |  0.5  |    0.79     |  0.69  |  0.41  |     **3**     | 0.24  | 0.45  |    0.76     |  0.80  |  0.56  |
> |  **30**   | 0.11  | 0.23  |    0.50     |  0.58  |  0.45  |     **4**     | 0.13  | 0.25  |    0.45     |  0.61  |  0.74  |
>
>
>
> #### **Comparison to RHP.**
>
> Our experiments suggest that transfer attacks using RHP are much weaker than the generator-based GAP [10] baseline. This is attributed to the RHP framework employing a severely under-fitted gradient transformer module using just a single convolutional layer. Nevertheless, we report below the results of RHP and will include these results in the revised version.
>
>
>
> | Generator  Training | Discriminator | VGG16        | ResNet152    | Inception-v3 | DenseNet121  | SqueezeNet   |
> | ------------------- | ------------- | ------------ | ------------ | ------------ | ------------ | ------------ |
> |                     |               |              | RHP  / Ours  |              |              |              |
> | ImageNet            | ResNet152     | 65.4  / 99.1 | 48.3  / 99.7 | 51.2  / 72.2 | 53.6  / 98.7 | 76.5  / 88.0 |
> |                     | Inception-v3  | 61.4  / 98.7 | 40.5  / 89.6 | 47.6  / 99.6 | 48.2  / 95.9 | 68.5  / 91.3 |
> | Comics              | ResNet152     | 62.8  / 99.1 | 48.5  / 99.7 | 51.5  / 71.6 | 54.5  / 98.5 | 76.3  / 86.9 |
> |                     | Inception-v3  | 64.5  / 99.0 | 38.2  / 90.7 | 50.5  / 99.7 | 47.6  / 96.8 | 72.4  / 93.2 |
> | Paintings           | ResNet152     | 54.3  / 98.1 | 45.3  / 99.7 | 48.3  / 58.8 | 50.2  / 97.1 | 68.3  / 81.7 |
> |                     | Inception-v3  | 49.4  / 98.6 | 38.2  / 88.7 | 50.0  / 99.4 | 48.7  / 95.6 | 68.4  / 90.8 |
>
>
> **Table 3 Results**.
>
> This is an interesting observation. Note that, for all results in Table 3, the generator was trained with ChestXNet as the discriminator, following the same protocol as that in CDA [7].  As can be observed from Table 3, the generator is more effective when trained with the Comics and Paintings datasets than with ChestX. This is explained by the fact that the ChestX dataset presents a larger domain shift, without any information about objects and color, which makes it hard for the generator to generalize to ImageNet-trained models at test time.
>
> #### **Layer Selection.**
>
> We agree that the choice of layer affects our results. However, we empirically found that the best-performing layer is independent of the target model to attack. As shown in Figures 3 and 4 in the supplementary material, the chosen layer to attack consistently transfers better than other layers in all settings at test time. In practice, the attacker can therefore choose the layer by simply validating once on a model different from the source architecture; there is no need to search for the best layer for every target model. We would also like to clarify that, for all models, we probed on average 5 layers, spanning from the input layer to the final classification layer. In other words, we do not require an exhaustive sweep of all layers.
>
>
> Regarding the axes for the plots in Figures 3 and 4, the y-axis corresponds to the transfer rate, and the x-axis indicates the model's layer. For ResNet models, the architecture typically contains 4  blocks, thus explaining the values on the x-axis. For example, ResNet152 contains four blocks of depth 3, 8, 36, and 3, and we chose layer _l_ to be at the end of each block. We will clarify this.

---

### Official Review · Reviewer_cUgt · 2021-07-16

**Rating:** 6
**Confidence:** 4

**Summary:**

The work studied the generalization of adversarial attacks that are induced by generative methods. Specifically, the transferability of the perturbations for different testing conditions are evaluated. The mid-level features of DNNs are used to learn a transferable perturbation generator. Evaluations of transferable attacks are performed on tasks with a different network architecture, and/or data distributions.

**Ethical Concerns:**

N/A.

**Limitations And Societal Impact:**

The authors mentioned their limitations pretty well, in terms of only testing on a single layer. However, it would be great if the authors could add additional experiments comparing with meta learning or domain randomization based attack for improving generalization of attacks.

**Main Review:**

originality: the work presents an interesting finding that by maximizing the difference of mid level features of neural networks between the input image and the perturbed image, the generated perturbation can be transferred to another task setting such as a different data distribution or a different network structure. This finding itself is a useful contribution. However, the method of using generative method to generate perturbation is not new. The results indicate that there is a large gap between the white-box setting and the black-box setting. This means that when the target data distribution is hugely different and/or target network architecture is different, the transferability of the attack degrades.

quality: the paper is technically sound and the claims are solid. However, the results indicate that there is still a large gap between the non-transfer case and the transfer case. In order to train an attacker that can transfer or generalize well to different settings, there could be other methods such as using meta learning to train the attacker on multiple data sources or multiple network architectures. It would be great if the author could provide results on meta learning based attack to see how good is the proposed attack comparing with meta learning based attack.

clarity: the paper is presented in a clear way except for some tiny fonts in some of the figures.

significance: the results are important and the insight from the paper is useful for other applications.

**Time Spent Reviewing:**

1.5

---

> ### Author Response · Authors · 2021-08-10
> **Response**
>
>
> Thank you for carefully reading our paper and for your encouraging remarks.
>
> #### **Meta-learning based attacks**
>
> Thank you for the suggestion. While we understand the ideas of attacking an ensemble of source models and of using multiple data sources, it is unclear to us what the reviewer has in mind in terms of "meta-learning". We would be happy to evaluate such an approach if you could give us more detail. In any event, below, we report the fooling rates with $\epsilon=10$ for an ensemble and for a multi-dataset case.
>
> |    Generator Training    | Discriminator at training time | VGG16 | ResNet152 | Inception | DenseNet121 | SqueezeNet | Average |
> | :----------------------: | :----------------------------: | :---: | :-------: | :-------: | :---------: | :--------: | ------- |
> |                          |                                |       |   Ours    |           |             |            |         |
> |       ImageNet10K        |           SqueezeNet           | 96.2  |   76.2    |   72.8    |    89.2     |    99.6    | 86.8    |
> |       ImageNet10K        |             VGG16              | 98.8  |   57.3    |   37.4    |    75.5     |    79.2    | 69.6    |
> |       ImageNet10K        |       SqueezeNet + VGG16       | 94.8  |   74.4    |   70.3    |    87.4     |    99.6    | 85.3    |
> |        Comics10K         |           SqueezeNet           | 96.4  |   79.2    |   75.2    |    90.5     |    99.7    | 88.2    |
> |       Paintings10K       |           SqueezeNet           | 95.4  |   73.7    |   71.5    |    87.9     |    99.6    | 85.6    |
> | Comics10K  +Paintings10K |           SqueezeNet           | 96.8  |   78.1    |   71.7    |    89.6     |    99.8    | 87.2    |
>
> When attacking an ensemble of networks during training, such as VGG16 and SqueezeNet, the average transfer rate is of 85.3%, which is closer to that of the best performing model among the considered two, that is, SqueezeNet with a fooling rate of 86.8%.
>
> We observe a similar trend when using multiple data sources to train the generator. For instance, by using both Comics10K and Paintings10K, we obtain an average fooling rate of 87.2%, which does not surpass that of each individual data source but remains close to the 88.2% reached when using Comics10K only.
>
> Both the experiments suggest that adding more data sources or more networks during training does not necessarily improve the transfer rates. We will include these results in the revised paper.

---

> > ### Comment · Reviewer_cUgt · 2021-09-02
> > **Response**
> >
> > Thanks for the authors for the response. I think the further results help to make the claim clearer but I think the idea is not that novel so I keep my rating.

---

### Official Review · Reviewer_9Cvd · 2021-07-16

**Rating:** 6
**Confidence:** 4

**Summary:**

The paper proposes an adversarial attack generated by another deep neural network (DNN), that shown to have potential to transfer the attack to different network architectures, different datasets and different tasks. The authors inspire the attack from an observation that mid-level features of a DNN share similarities with different architectures, datasets and tasks. Learning the adversarial perturbation generator network to attack the mid-level features of a DNN shown to transfer to different attack settings. The effectiveness of the attack is justified on different attack settings (white box and variants of black box attacks) that emphasize the transferrable nature of the attack.

**Limitations And Societal Impact:**

I appreciate the authors for covering the limitations and broader impact of their work.

**Main Review:**

Originality:
Learning the generator to attack the mid-level features and demonstrating such generator to transfer to different attack settings is interesting and novel. Previous works are cited and differences are discussed sufficiently.

Quality:
The paper provides an interesting finding and technically sound. The proposed method is appropriate and the claims are well supported by the extensive experimental results. This is a complete work and the strengths of the method are reveled by crafting the experiments carefully. The weakness of the method is not sufficiently addressed (see below).

Clarity:
The paper is clearly written, well organized and also provided details to reproduce the results.

Significance:
The paper addresses an important topic of transferring the attack to different test settings.
The results of the proposed attack outperforms the previous attacks in all the different attack settings. Future works would benefit from the unique findings made in this work.

I appreciate that authors study the effectiveness of their method on few of the adversarially trained models. However, I wonder that the authors missed the PGD based adversarially trained model as a baseline to test their method. CK Mummadi et al. has shown that PGD based adversarially trained models are also robust to universal perturbations and thus raises a question how their method performs on such defense method.

Besides the interesting finding that mid-level features are shared across different architectures, my major concern about this work is the assessment of the proposed attack against potential defenses. How strong is the proposed attack when a defense is crafted to address it specifically? It seems the attack objective could be easily get integrated into PGD based adversarial training to achieve a robust model. If so, I wonder whether the proposed attack would circumvent such defense mechanism. I believe this is an important discussion to be addressed in order to evaluate the significance of the proposed attack. Finally, I rate this paper as marginally above acceptance threshold. However, I am willing to reconsider my rating based on the author responses and other reviews.

CK Mummadi et al.  Defending Against Universal Perturbations With Shared Adversarial Training, ICCV 2019.

-----------------------------------
Post rebuttal remarks:
I read the other reviews and authors responses. I thank the authors for conducting additional experiments on defended models. The results show that the proposed attack is effective against HGD and R&P defended models, but not effective against PGD and Feature Denoising based defended models (same as other attacks). However, my major concern of crafting a defense that specifically target to circumvent this attack still remains open and hence I keep my rating same as before.

**Time Spent Reviewing:**

4

---

> ### Author Response · Authors · 2021-08-10
> **Response**
>
> Thank you for your careful reading of our paper and your insightful feedback.
>
> #### **Attacking Defense Models**.
>
> Following the reviewer's suggestion, we evaluate our approach on models adversarially trained with PGD [a] and with 3 other defenses including high-level representation guided denoiser (HGD) [b], input transformation through random resizing and padding (R&P) [c], feature denoising adversarially trained ResNet-152 (FD) [d].  HGD [b] and R&P [c] were the rank-1 and rank-2 submissions in the NeurIPS 2017 defense competition, respectively, and FD [d] was the rank-1 submission in the Competition on Adversarial Attacks and Defenses 2018.
>
> As for Table 4 in the main paper, we use a naturally-trained Inception-V3 to generate adversarial examples with $\epsilon$=16 and report the percentage of increase in top-1 error.
>
> |         Method         | HGD  | R&P  | PGD  |  FD  |
> | :--------------------: | :--: | :--: | :--: | :--: |
> | Error on clean samples | 19.4 | 20.2 | 53.8 | 53.5 |
> |          RHP           | 26.8 | 23.3 | 2.4  | 2.4  |
> |          CDA           | 49.5 | 14.6 | 3.6  | 4.1  |
> |          Ours          | 56.2 | 38.2 | 3.3  | 4.3  |
>
> From these results we observe that our attack is more effective than CDA in breaking HGD and R&P defenses, whereas it performs on par with it for PGD and FD defenses, with none of the methods successfully attacking these models.  Note, however, that with PGD and FD, the error rate on the clean samples is significantly higher than with other defenses, making these strategies ill-suited for practical applications.
>
>
> [a] Madry, A., Makelov, A., Schmidt, L., Tsipras, D., Vladu, A.: Towards deep learning models resistant to adversarial attacks. In: ICLR (2018)
>
> [b] Liao, F., Liang, M., Dong, Y., Pang, T., Hu, X., Zhu, J.: Defense against adversarial attacks using high-level representation guided denoiser. In: CVPR (2018)
>
> [c] Xie, C., Wang, J., Zhang, Z., Ren, Z., Yuille, A.: Mitigating adversarial effects through randomization. In: ICLR (2018)
>
> [d] Xie, C., Wu, Y., Maaten, L.v.d., Yuille, A.L., He, K.: Feature denoising for improving adversarial robustness. In: CVPR (2019)

---

### Official Review · Reviewer_gZcr · 2021-07-17

**Rating:** 6
**Confidence:** 4

**Summary:**

This paper studies the transferability of adversarial examples crafted by attacking one model to another model. The main idea is to employ a generator to produce adversarial examples and maximize feature divergence on a latent space.

**Limitations And Societal Impact:**

Although the experiments are quite comprehensive, the proposed method is restrictively novel. Moreover, the arguments of the proposed method are too general, hence needing a more comprehensive study.

**Main Review:**

The core idea of this work is to run an attack on mid-features at the intermediate layer of the source network. The main intuition is that mid-layers tend to learn common patterns (e.g., textures) across various deep nets trained on the same dataset or similar datasets.

However, the arguments in the paper regarding why maximizing feature divergence on mid-layers can boost transferability have not been addressed comprehensively and thoroughly. The current arguments are too general to me. It would be better if the authors delve into what really happens with mid-layer features and what changes in the pattern of adversarial examples when maximizing feature divergence on mid-layers. This would help insightful demonstrate the usefulness of the proposed method.

--Post rebuttal---
The responses from the authors have addressed my concerns so I have increased my score.


**Time Spent Reviewing:**

2

---

> ### Author Response · Authors · 2021-08-10
> **Response**
>
> We thank you for your feedback and address your concern below.
>
> #### **Analysis Thoroughness**.
>
> Our work explores the reasons for transferability across architectures, data, and tasks. Our core finding is that the mid-level filter banks of different deep networks bear strong similarities. We then exploit this to train a generator using a feature separation loss, and show that this outperforms the baselines GAP and CDA by a large margin in Tables 1, 2, 3, and 5.
>
> We therefore respectfully disagree with the reviewer that our analysis is not sufficiently thorough. Beyond the quantitative results supporting the transferability of our approach, we provide several visualization means that further strengthen our arguments. In particular, in Figures 1 and 2 of the main paper, we visualize the filters of different architecture and for different task, observing a strong correlation between the perturbation pattern and filter representations; in Figures 3 and 7 of the main paper and Figures 1 and 2 of the supplementary material, we provide t-SNE plots to visualize features at different layers, for different attack strategies and different models; in Figure 4 of the main paper, we analyze the top activated channels for different datasets; in Figures 3 and 4 of the supplementary material, we analyze the influence of the choice of layer on transferability. Furthermore, following the suggestion of Reviewer WvGo, we provide CKA values to quantify the similarity between different layers of different architectures. Specifically, we observed that the CKA values are typically above 0.75 for the intermediate layers that we considered in our experiments, further explaining the reason for high transferability. For example, on transfer attacks between VGG16 and DenseNet121, the CKA values are highest between the intermediate layer 8 in DenseNet121 and layers 18 and 23 in VGG16. Similarly, between ResNet152 and DenseNet121, the CKA value is 0.76 for the optimal generator configuration, which relies on layer 3 in ResNet152 and layer 8 in DenseNet121. We discuss this in the revised paper.
>
> |           |       |       | DenseNet121 |        |        |               |       |       | DenseNet121 |        |        |
> | :-------: | :---: | :---: | :---------: | :----: | :----: | :-----------: | :---: | :---: | :---------: | :----: | :----: |
> | **VGG16** | **3** | **6** |    **8**    | **10** | **12** | **ResNet152** | **3** | **6** |    **8**    | **10** | **12** |
> |   **4**   | 0.71  | 0.64  |    0.26     |  0.21  |  0.14  |     **0**     | 0.69  | 0.54  |    0.25     |  0.2   |  0.13  |
> |   **9**   | 0.48  | 0.58  |    0.42     |  0.30  |  0.19  |     **1**     | 0.74  | 0.82  |    0.42     |  0.33  |  0.21  |
> |  **18**   | 0.31  | 0.58  |    0.75     |  0.58  |  0.34  |     **2**     | 0.57  | 0.83  |    0.65     |  0.49  |  0.31  |
> |  **23**   | 0.26  |  0.5  |    0.79     |  0.69  |  0.41  |     **3**     | 0.24  | 0.45  |    0.76     |  0.80  |  0.56  |
> |  **30**   | 0.11  | 0.23  |    0.50     |  0.58  |  0.45  |     **4**     | 0.13  | 0.25  |    0.45     |  0.61  |  0.74  |
>
>
> We could of course provide additional examples of such analyses, but they would simply show the same behavior as what can be observed from the current figures. We therefore hope that the reviewer can re-assess our work, or provide us with more specific feedback regarding the kind of analysis that they would like us to perform so that we can address their concern.

---

> > ### Comment · Reviewer_gZcr · 2021-08-22
> > **More discussions**
> >
> > Thanks for your rebuttal.
> >
> > Actually, for this kind of simple and practical work, I expect a deeper explanation of visualization. My questions are as follows.
> > - In Figure 2, you use the technique of [12], which maximizes the mean activation of each filter, for visualization of the low-level, mid-level, and high-level filters. Basically, this will create a synthesized image that maximizes the average of neurons on the feature map corresponding to this filter. My question is how those synthesized images relate to adversarial examples? What is the connection and correlation between crafted adversarial examples and those synthesized images? Why does crafting adversarial examples which maximize the divergence with their benign images on mid-layers help to transfer better? What is the relation between maximizing the divergence with their benign images on a layer and the pattern of synthesized images? I believe that the current argument of that low-level layers extract color and edge information, high-level layers are more focused on the object representation, and the mid-level filters learn more nuanced features, such as textures are too general. I cannot explicitly see how they link to the process of learning adversarial examples by maximizing representation divergence.
> > - Figure 3 is less informative to me.  On layer 4, although we can see more overlapping between ImageNet and painting, by visualizing using tSNE, we project multi-dimensional representations to 2D space, hence projected representations are distorted and they might not overlap on the original space. In addition, except for chestx, other representations always stay closely regardless of layers.
> >
> > I acknowledge the comprehensive experiments conducted. However, I believe that I need more explanations of how the synthesized images obtained by [12] relate to crafted adversarial examples and how crafted adversarial examples by maximizing representation divergence activate neurons of the corresponding neuron so that I can increase my score.

---

> > > ### Author Response · Authors · 2021-08-24
> > > **Re: "More discussions"**
> > >
> > > Thank you for your additional feedback and for giving us a chance to clarify our work. We answer your questions below, albeit in a slightly different order than that they were asked in.
> > >
> > > - What is the relation between maximizing the divergence with their benign images on a layer and the pattern of synthesized images? How do those synthesized images relate to adversarial examples? What is the connection and correlation between crafted adversarial examples and those synthesized images?
> > >
> > > Recall that filters in DNN encode different types information, and a filter activation with a large magnitude implies the presence of a specific pattern in the input image. When maximizing the feature separation loss of Eq. (1), the generator cannot disrupt all filters because it is inconceivable for a single image to activate all filters with a large magnitude. Therefore, it learns to disrupt only a small subset of filters. To evidence this, we plot the shift in activation values for the 100 filters that have the largest shift between normal and adversarial images using 500 random images at [https://anonymous.4open.science/r/TransferablePerturbations-77BB/2.Filters_shift_in_activations_boxPlots/readme.md](https://anonymous.4open.science/r/TransferablePerturbations-77BB/2.Filters_shift_in_activations_boxPlots/readme.md). These plots show that the adversarial images disproportionately activate only a subset of the filters by a large magnitude.
> > >
> > > Furthermore, as shown in Figure 1, the adversarial images from the output of the generator, such as the boat image in the top block and the cat and train images in the bottom block, contain wavy texture patterns (please zoom in onto these images to see this; we will make this more visible in the final version). These texture patterns visually resemble the images synthesized using [12] corresponding to the filters that have the largest shift in activation values between normal and adversarial images. We anonymously provide clearer visualizations of unbounded adversaries and final bounded adversarial images with $\epsilon=10$, together with synthesized images corresponding to the most disrupted filters at [https://anonymous.4open.science/r/TransferablePerturbations-77BB/1.Adversarial_Images_and_Filter_Synthesised_images/readme.md](https://anonymous.4open.science/r/TransferablePerturbations-77BB/1.Adversarial_Images_and_Filter_Synthesised_images/readme.md). These figures highlight a strong connection between unbounded adversarial images and synthesized images corresponding to the most disrupted filters.
> > >
> > >
> > > - Why does crafting adversarial examples which maximize the divergence with their benign images on mid-layers help to transfer better?
> > >
> > > As discussed in our previous answer, the most disrupted filters strongly correlate with the perturbation patterns in the adversarial images. Furthermore, as can be seen in the figures provided anonymously in our previous answer, different architectures have similar mid-level filters. Thus, the set of filters disrupted by the loss in Eq.(1) is common across architectures, which facilitates the transfer.
> > >
> > > We provide additional visualizations of adversarial images obtained by attacking different layers, ranging from the bottom input layer to the final classification one, for all networks at [https://anonymous.4open.science/r/TransferablePerturbations-77BB/3.Adversarial_images_vs_layer_position/readme.md](https://anonymous.4open.science/r/TransferablePerturbations-77BB/3.Adversarial_images_vs_layer_position/readme.md). We consistently observe that the bottom layers, encoding mostly color and edges, are not effective in generating adversaries because they require larger perturbation strength. Furthermore, attacking the top layers produces adversarial images that have more object-like patterns probably overfitting to the decision boundaries of the source networks. By contrast, the mid-layers focus on textures that are common across architectures, and thus attacking these features yields effective transfer.
> > >
> > >
> > > - Clarification about Figure 3
> > >
> > > We understand the reviewer's concerns and will revise the discussion of this figure accordingly. Our goal with Figure 3 was to analyze the relative domain shift between datasets and the reason for the transferability of our attacks across such datasets. In short, the plots show a larger domain shift between ChestX and ImageNet than between Paintings/Comics and ImageNet, which correlates with the slightly lower transfer rates reported in Table 2.

---

> > > > ### Comment · Reviewer_gZcr · 2021-09-01
> > > > **Thanks for your comprehensive feedback**
> > > >
> > > > Dear authors,
> > > >
> > > > Thanks for your meaningful and comprehensive feedback which addressed my concern. I will increase my score to 6. I encourage you to incorporate those visualizations and explanations to the paper because it is valuable to the community.

---

> > > > > ### Author Response · Authors · 2021-09-01
> > > > > **Re: "Thanks for your comprehensive feedback"**
> > > > >
> > > > > Thank you for your encouraging comments. We will definitely incorporate the additional visualizations and explanations in the final version.

---

### Decision · Program_Chairs · 2021-09-27

**Decision:**

Accept (Poster)

**Comment:**

The paper studied the generalization of adversarial attacks induced by generative methods. The authors found that, by maximizing the difference of mid-level features of neural networks between the clean and perturbed images, the generated perturbation can be better transferred to another task setting such as a different data distribution or a different network structure. The finding is interesting and useful to the community. However, there is still a gap between the attack performance under the white-box and black-box settings. Furthermore, the proposed attack is effective against HGD and R&P defenses, but not effective against PGD and feature-denoising-based defenses. We suggest the authors discuss these in more detail in future revisions.